# Utility and cost-effectiveness of LiverMultiScan for MASLD diagnosis: a real-world multi-national randomised clinical trial

## Abstract

**Background:** Increasing prevalence of metabolic dysfunction–associated liver disease (MASLD) and metabolic dysfunction–associated steatohepatitis (MASH) poses a growing healthcare burden. Noninvasive diagnostic tools to replace liver biopsy are urgently needed. We investigated the utility and cost-effectiveness of including multiparametric magnetic resonance imaging (mpMRI) to the management of adults with suspected MASLD multi-nationally.

**Methods:** RADIcAL-1, a 1:1 randomised controlled trial (standard-of-care [SoC] *vs*. imaging arm [IA; SoC+mpMRI]) included 802 participants from Germany, Netherlands, Portugal and UK. Wilcoxon-rank tests were used to compare access to healthcare practitioners, patient assessments and proportion of patients with a diagnosis (%diagnosis). Liver fat and disease activity (corrected T1 [cT1]) were used to identify patients not requiring biopsy in the imaging arm. Primary endpoint was mpMRI cost-effectiveness and improvement in resource use (visits avoided) using mpMRI.

**Results:** mpMRI is cost-effective with an ICER of €4968/QALY gained. 403 were randomised to IA and 399 to SoC. SoC has significantly more specialist appointments ($p = 0.015$) and patient assessments ($p < 0.001$). Across all involved hospitals, %diagnosis is significantly higher in the imaging arm ($p = 0.0012$). cT1 correctly classifies 50% of patients without MASH with fibrosis and can avoid biopsy. Including all costs, the imaging arm incurs higher short-term per-patient healthcare expenditure compared to the SoC arm (€1,300 *vs*. €830).

**Conclusion:** Adding mpMRI to SoC for the management of adults with suspected MASLD multi-nationally is cost-effective, enhances rate of diagnosis multi-nationally and increases rate of diagnosis without increasing other liver-related health care resource use. Due to the need for standardisation of SoC, widespread use can support optimisation of the MASLD clinical pathway and improve long-term patient management.

## Plain language summary

Steatotic liver disease is a global health problem which needs better diagnostic pathways. Here, we compared the number of doctor visits, the speed of diagnosis, and whether the cost of adding an MRI scan (LiverMultiScan) is justified by the improvement in patients' quality of life across Germany, the Netherlands, Portugal, and the UK. Findings show that using an MRI scan is a safer and pain-free alternative that can help doctors diagnose more people with fewer visits, making it a cost-effective option. These results are important because they show that using the MRI scan is affordable and effective enough to be recommended as it can make diagnosing liver disease faster, more accurate, and less invasive.

Metabolic dysfunction–associated liver disease (MASLD) encompasses a spectrum of disease ranging from fatty liver (simple steatosis) to metabolic dysfunction–associated steatohepatitis (MASH) without fibrosis, MASH with fibrosis, and MASH with cirrhosis. In Europe, patients with MASH experience up to 174,564 disability-adjusted life years (DALYs)[1] and the annual cost of management is estimated to be >€35 billion in direct costs.

However, despite this, the drivers of the economic burden are not well researched[2]. The reference standard for assessing the severity of MASLD is liver biopsy, which is costly, invasive, and has inherent limitations including inter- and intra-sample variability, pain, bleeding, and infection risk[3]. Alternate diagnostic approaches using noninvasive testing (blood-based markers) and liver stiffness measurement (using transient elastography)

✉ e-mail: elizabeth.shumbayawonda@perspectum.com

also have other limitations, including specificity (blood-based markers) to support disease stratification and monitoring, and high variability and operator dependency (semi-quantitative ultrasound assessments). The application of sequential investigation is the current approach to clinical management[4] and patient stratification. However, the concurrent use of multiple sub-optimal diagnostic tools leads to greater resource allocation and conflicting results impacting clinical conclusions, which can lead to misdiagnosis or underdiagnosis within the management of MASL[5]. Despite ongoing work to support patient management, in the UK[1] for instance, an estimated 80% of the prevalent MASH population remain undiagnosed, a statistic which poses a great economic impact on healthcare resources in the long term. To support early identification, stratification, and intervention for those with suspected MASLD, at highest risk of disease progression and adverse clinical outcomes, there is a need for a more streamlined, cost-effective, objective (quantitative), noninvasive diagnostic approach.

The European Association for the Study of the Liver (EASL) clinical practice guidelines for the management of MASLD recommend screening of patients at risk of MASH to avoid adverse clinical outcomes[4] including cardiovascular disease[6], coronary artery disease, arrhythmia, and stroke. To date, MASLD stratification focuses on identifying advanced fibrosis (F3-F4), however, targeting those individuals with MASH (particularly those with fibrosis) facilitates implementation of the most appropriate and effective interventions[4] including lifestyle modification or potential MASH pharmacotherapy (when available).

There are a range of currently available noninvasive markers of fibrosis including magnetic resonance elastography (MRE), shear wave elastography (SWE) and vibration controlled transient elastography (VCTE)[7]. However, the stiffness measurements (kPa) are not directly comparable between each other and there is a lack of consensus on cut-off values for differentiating degree of stiffness, especially across different populations[4,7] (especially in the presence of comorbidities such as diabetes). This leads to low repeatability of these technologies and limits their monitoring capabilities[8]. Moreover, as these techniques measure liver stiffness (as a surrogate of fibrosis), they have been shown to be less sensitive in detecting MASH[4]. Multiparametric magnetic resonance imaging (mpMRI), a standardised non-invasive non-contrast technology that uses post-processing of MRI images[9], directly characterises disease activity (using corrected T1 [cT1] a metric of disease activity including fibro-inflammation), liver fat (using proton density fat fraction [PDFF]), and liver iron (using T2*)[10–12]. Studies have demonstrated the improved ability of cT1 in identifying and stratifying steatotic liver disease when compared to other noninvasive technologies[8,13,14] (including vibration-controlled transient elastography [VCTE], magnetic Resonance Elastography [MRE] and sheer wave elastography [SWE]), in predicting clinical outcomes[14,15] and in monitoring and detecting early responses to treatment in those with MASLD/MASH[16,17]. In terms of sequential testing alongside other techniques, Blake and colleagues have shown that mpMRI, used as an adjunct or alternative to VCTE, may lead to significant cost savings in the MASLD diagnostic pathway[18], whilst Schaapman et al.[19] showed the prognostic utility of mpMRI to support risk assessment alongside other techniques in patients suspected of MASLD[19].

The primary objective of this randomised control trial was to investigate whether the introduction of mpMRI as a standardised diagnostic test for liver disease can prove a cost-effective method by reducing the number of patients with suspected MASLD who incur liver-related consultations, including liver biopsies. Therefore, the aim of this study was to evaluate the utility and cost-effectiveness of using mpMRI to risk stratify adults with suspected MASLD in a multinational setting.

Our findings show that multiparametric MRI is a cost-effective tool (ICER/QALY: €4929) that improves diagnosis rates, reduces the need for specialist consultations, and avoids unnecessary biopsies, offering a safer, pain-free alternative without increasing liver-related healthcare use. Widespread adoption of mpMRI optimises the MASLD clinical pathway,

enhancing patient management by improving clinical efficiency and reducing long-term MASLD care costs through streamlined management.

## Methods
### Study design
RADIcAL-1, a prospective, randomised controlled, multi-national trial, included patients recruited from 10 clinical centres in: Ulm (Germany), Leiden (Netherlands), Coimbra (Portugal) and 7 sites across the UK[20]. European sites were set up in 2017 (with recruitment continuing up until 2020 in Ulm and Leiden) while UK sites joined the study at varying times between 2019 and 2020. Patients were randomised in a 1:1 fashion (using a random combination of the inclusion criteria) to one of two study arms: standard of care (SoC) *versus* imaging arm (standard care with the addition of multiparametric MRI [mpMRI]). Figure 1 summarises the study design and trial profile[20]. The intended patient follow-up period was 12 months, however, due to the Covid-19 pandemic, some sites shortened the follow up duration to 6 months (minimum follow-up period). The trial was registered as a clinical investigation (NCT03289897) and principals of Good Clinical Practice and those of the 1975 Declaration of Helsinki were observed.

Patients gave written informed consent to participate in the study. The research ethics committees at each host institution provided granted approval in Ulm (198/17), Leiden (P17.076), Coimbra (CE-030/2017), and the UK (18/SC/0725) for the protocol, patient facing documents (informed consent form, participant information sheet) and all proposed advertising materials[20]. The study enrolment ran from 5 May 2017 to 31 December 2020 and is now completed.

### Patient recruitment and patient randomisation
To be included in the trial, all participants had to be aged 18-75 years and be due to undergo evaluation for suspicion of MASLD (Fig. 1). For inclusion patients had to have either elevated liver enzymes, imaging suggestive of fatty liver disease, or have the presence of 3 or more of the following criteria: insulin resistance or type 2 diabetes mellitus, obesity, hypertension, elevated triglycerides, or low HDL-cholesterol. Exclusion criteria included contraindication to MRI, proven liver disease other than MASLD, liver transplantation, clinical signs of chronic liver failure, pregnancy, alcohol overuse/abuse and any other cause of disease (such as autoimmune liver disease, cirrhosis, Wilsons disease, cystic fibrosis-related liver disease, viral hepatitis, etc.), which in the opinion of the investigator, based on medical records, may affect the participant's ability to participate in the study. The full study design for this trial has been reported by Tonev et al.[20] and full inclusion and exclusion criteria are shown in Supplementary table 1.

### Study design and Heterogeneity between study centres
This study was a real-world evidence (RWE) prospective study. Healthcare systems differ across countries, and thus, patients who met the inclusion criteria were recruited into this study from various settings. In Ulm, patients were primarily recruited from MASLD secondary care referral clinics, while in Leiden they were recruited from specialized outpatient clinic for patients with suspected MASLD at a tertiary referral centre. In Coimbra, patients were recruited from diabetes general practitioner specialist clinics and in UK they were recruited from MASLD or diabetes secondary care clinics. Therefore, to enable evaluation of the true benefit inclusion of a new technology would have on SoC, care across centres was not homogenised (mandated to a specific hypothetical pathway) but rather reflected real-life clinical practice. Consequently, evaluation and data collected reflected the variable treatment and management pattern which depended on clinician discretion and the standard of care (SoC) and standard of practice (SoP) at each included site[21]. As a result, diagnosis was dependent on the centres SoC/SoP and clinician discretion.

### Health care resource use, classification of liver-related health-care resource use and rate of diagnosis
To quantify the types and frequency of health care resource use following randomisation, participants completed a paper-based health care resource

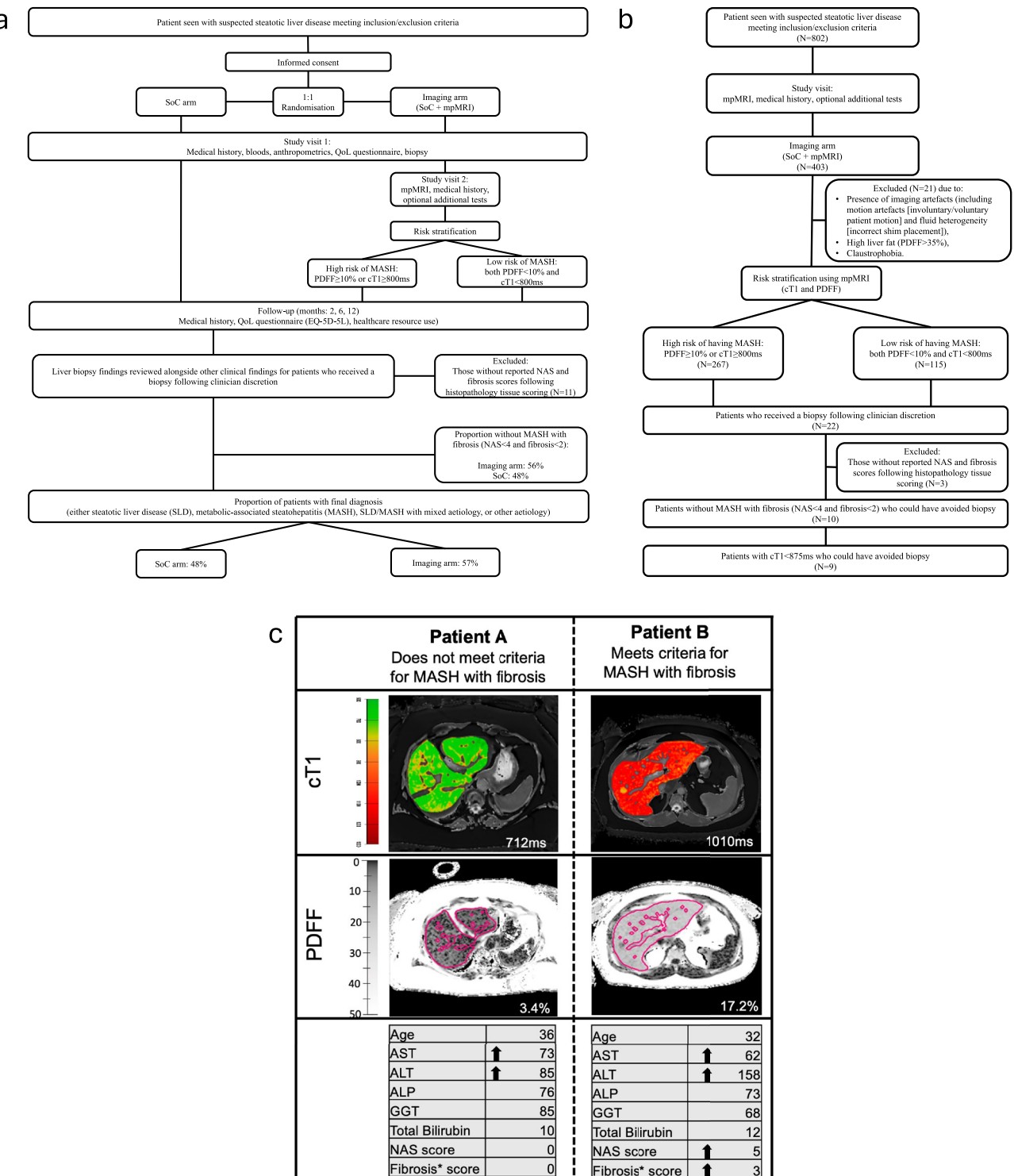

**Fig. 1 | Summary of study visits for all participants and the RADIcAL1 trial profile including a case study for two female patients with suspected MASLD.** Summary of study visits for (**a**) all participants and (**b**) RADIcAL1 trial profile. Also shown in (**a**) is the proportion of patients who received a biopsy following clinician discretion but did not have Metabolic dysfunction-associated steatohepatitis (MASH) with fibrosis (MAS < 4 and fibrosis <2), and those who received a final diagnosis in both the standard-of-care (SoC) and imaging arms, and (**b**) the study visits for only those randomised into the imaging arm (SoC + mpMRI), the proportion of patients classified as high risk using mpMRI (PDFF ≥ 10% or cT1 ≥ 800 ms), those who received a liver biopsy and those with cT1 < 875 ms who could have avoided a liver biopsy to assess for MASH with fibrosis. (**c**) cT1 and PDFF maps for two female patients with suspected Metabolic dysfunction–associated steatotic liver disease (MASLD), both with BMI 33, no diabetes, with elevated liver enzymes (ALT and AST) who underwent liver biopsy following clinician discretion to confirm diagnosis. Arrows indicate elevated biochemical markers (above the upper limit of normal) and histological stage. In the cT1 maps, lower values (cooler colours) represent areas with lower cT1 values, and therefore, lower disease activity, whereas higher cT1 values (warmer colours) represent areas of the liver with higher disease activity. In the PDFF maps, darker shades represent lower liver fat, whilst lighter shades represent higher liver fat. BMI body mass index, ALT alanine aminotransferase, AST aspartate aminotransferase; cT1 corrected T1, PDFF proton density fat fraction, mpMRI multiparametric MRI.

use questionnaire (HCRUQ) at 2-, 6- and 12-months following inclusion into the trial[20]. HCRUQs collected data on outpatient investigations including blood tests, biopsies, medical imaging, and medical consultations. The HCRUQs did not specifically stipulate recall of *liver-related* health care. Thus, to infer as best as possible if the health care was liver related, three clinicians (expert hepatologists, each with over 20 years' experience) were blinded to assess the nature of the health care using an inclusive mapping methodology. A five-level categorisation was applied, and a binary classification used to map if the healthcare was liver related (A) not-liver related (B) as follows:

$$Definitely\ liver-related \rightarrow A, Probably\ liver-related \rightarrow A, Possibly\ liver\ related \rightarrow A,$$

$$Probably\ not\ liver-related \rightarrow B, Definitely\ not\ liver-related \rightarrow B$$

In this investigation, rate of diagnosis (%-with-diagnosis) was assessed as the proportion of patients with a final diagnosis by the end of the trial follow-up period.

### Cost of health care resource use

Unit health care costs from a health system payer perspective (in UK, Germany and Portugal: a national perspective[22–25], and in Netherlands the hospital local costs) were sourced and multiplied by the frequency of use for each patient (supplementary table 1). Guidance on adjusting for inflation and currency changes within economic studies was followed[26]. In the UK, nominal values were inflated to 2019/2020 real values using Office for National Statistics (ONS) health index for UK pound sterling, while the Organisation for Economic Cooperation and Development (OECD) Health consumer price index (CPI) were applied for Portugal, Germany and Netherlands[27,28] (supplementary table 2). Due to the short follow-up duration, costs and consequences were not discounted.

### Quantitative MR acquisition protocol and image analysis

The mpMR scanning protocol was installed, calibrated and both phantom and volunteer tested on all MRI scans at the included centres. All mpMR images were obtained in a 15-minute period using a non-contrast abdominal MRI scan following the LiverMultiScan (Perspectum Ltd, Oxford, UK) image acquisition protocol (standardised across major vendor scanners [Siemens, Philips, GE] and field strength [1.5 T/3 T])[9]. Four transverse slices obtained at the porta hepatis location in the liver were acquired for each participant using a shortened modified look-locker inversion (shMOLLI) and a multi-echo spoiled gradient-echo sequence to quantify T1, iron (T2*) and fat (PDFF) (supplementary table 3). Following the scan, anonymised MR data were analysed off-site by specialised imaging analysts trained in abdominal anatomy and artefact detection who were blinded to the clinical data. During image analysis, cT1 and PDFF maps of the liver were delineated into whole liver segmentation maps using a semi-automatic method. Non-parenchyma structures such as bile ducts and large blood vessels as well as image artefacts were excluded from image analysis. All MRI scans were read for incidental findings following local site standards of practice.

### Outcomes

The primary objective for this trial was to measure the cost effectiveness of LiverMultiScan as a standardised diagnostic test for liver disease in different EU territories. The primary outcome for this trial was to investigate the utility and cost-effectiveness of including multiparametric magnetic resonance imaging (mpMRI) to the management of adults with suspected MASLD incurring liver-related hospital consultations including liver biopsies between the study arms[20]. It is worth noting that, in this study, in line with definitions and thresholds reported in literature[29–31] an incremental cost-effectiveness ratio (ICER) of ≤£20,000 per QALY gained has been considered as cost-effective in this population. To take into account variation in the difference healthcare systems, a conservative approach was used and thus, an ICER of ≤£10,000 (~€11,200) per QALY gained was considered

as cost-effective in this study. There were multiple secondary objectives for this trial including assessment of %-with-diagnosis, frequency of diagnosis and resource use (including access to specialist care) prior to MASLD/MASH diagnosis. Moreover, the secondary outcomes included assessment of certainty of diagnosis, frequency of diagnosis and resource use (measured as rates of liver related outpatient investigations/ consultations/ hospital admissions during the study)[20].

### Sample size calculation

In a study by Blake et al.[18] the use of LiverMultiScan was found to result in a decrease in and 18% decrease in the use of biopsy. Adopting a conservative target of identifying a 14% decrease across different regions, each randomisation arm is required to have 402 patients to maintain statistical significance with more than 80% power and show a difference in proportion of patients having consultations between the 2 arms. Due to the size of the trial, assuming there is a 25% dropout rate a total cohort of 1072 patients with will need to be recruited into the trial[20].

### Statistics and reproducibility

For each study outcome, data analysis involves a between group comparison across the two trial arms. Continuous variables were reported as mean and standard deviation or median and range as appropriate, categorical variables were reported as frequency and percentage. Health care resource use results are reported as descriptive summary statistics.

Wilcoxon rank tests were used to compare the access to healthcare practitioners and patient assessments between imaging [SoC + mpMRI] and SoC arms. Two sample t-tests were used to investigate the difference in %-with-diagnosis between randomisation arms. For patients incurring a liver biopsy, only procedures that occurred at the study clinic/centre were considered to be related to their suspected diagnosis.

Following clinical guideline recommendations, a biopsy which could have been avoided (avoidable biopsy) was defined as one that did not meet the criteria for MASH with fibrosis (MASLD activity score [MAS] ≥ 4 and fibrosis ≥ 2). To investigate the utility of mpMRI metrics to stratify patients and to better understand the estimated resource use investigations were performed in the imaging arm where PDFF and cT1 were used to classify patients as either having a low (both PDFF < 10% and cT1 < 800 ms) or high (either PDFF ≥ 10% or cT1 ≥ 800 ms)[17] risk of having MASH. Furthermore, a cT1 value <875ms[26] was used to stratify patients to identify those who could have avoided the procedure.

Statistical analyses were performed using R version 1.4.1103, with $p < 0.05$ deemed to be indicative of statistical significance throughout.

### Patient and public Involvement

There was no involvement from patients or members of the public (patient and public involvement) in the design, or conduct, or reporting, or dissemination plans of the research presented herein.

### Reporting summary

Further information on research design is available in the Nature Portfolio Reporting Summary linked to this article.

## Results

### Participant demographics, randomisation, and baseline characteristics

Between 2017 and 2020, 802 patients were recruited across four countries: 216 in Ulm, 177 in Leiden, 154 in Coimbra and 255 in the UK (Table 1). Patients that experienced claustrophobia and could not have an MRI, though willing to still partake in the study and submit resource use questionnaires were included in the analysis. This approach enabled further implementation insights into care provided when the intended imaging was not possible. There was high patient acceptance of the MRI scan resulting with only 1.7% (7/403) declining or being unable to have an MRI scan.

403 patients were randomised into the imaging arm (IA) while 399 randomised into the SoC arm[18]. Randomisation was done automatically

**Table 1 | Overall and regional baseline demographic summary**

| Summary metrics | Total n = 802 | | | Ulm (Germany) n = 216 | | | Leiden (The Netherlands) n = 177 | | Coimbra (Portugal) n = 154 | | UK n = 255 | | | |
|---|---|---|---|---|---|---|---|---|---|---|---|---|---|---|
| Group | Imaging | SoC | P-value | Imaging | SoC | P-value | Imaging | SoC | Imaging | SoC | Imaging | SoC | P-value Imaging | P-value SoC |
| *Cohort description* | | | | | | | | | | | | | | |
| Number of patients, N (%) | 403 (50.2) | 399 (49.8) | | 107 (49.5) | 109 (50.5) | | 89 (49.7) | 88 (49.7) | 79 (51.2) | 75 (48.8) | 128 (50.0) | 127 (50.0) | <0.001 | <0.001 |
| Age, years (±SD) | 53.2 (13.3) | 53.0 (13.3) | 0.74 | 49.5 (13.8) | 49.8 (13.2) | | 51.0 (13.5) | 50.6 (13.7) | 60.9 (8.5) | 61.5 (9.8) | 53.2 (13.1) | 52.5 (12.9) | 0.36 | 0.39 |
| Sex (Male, N (%)) | 225 (55.8) | 228 (57.1) | 0.71 | 58 (54.0) | 61 (56.0) | | 52 (58.0) | 52 (59.0) | 38 (48.0) | 37 (49.0) | 77 (60.0) | 78 (61.0) | <0.001 | <0.001 |
| Weight, kg (±SD) | 89.9 (17.2) | 90.9 (18.7) | 0.91 | 87.2 (12.6) | 89.9 (18.3) | | 95.1 (19.4) | 90.7 (19.8) | 83.8 (15.7) | 81.6 (12.2) | 92.6 (17.9) | 95.8 (20.5) | 0.001 | <0.001 |
| BMI, % (±SD) | 31.1 (5.3) | 31.3 (5.7) | 0.91 | 29.2 (4.2) | 30.3 (6.1) | | 31.1 (5.2) | 30.4 (5.5) | 31.7 (5.5) | 30 (4.5) | 32 (5.7) | 32.9 (5.8) | <0.001 | <0.001 |
| T2DM, N (%) | 176 (43.7) | 164 (41.1) | 0.47 | 27 (25.0) | 34 (31.0) | | 25 (28.4) | 16 (18) | 55 (69.0) | 45 (60.0) | 69 (54.0) | 69 (54.0) | <0.001 | <0.001 |
| *Biochemical markers* | | | | | | | | | | | | | | |
| ALT, IU/L (±SD) | 53.4 (32.3) | 56.7 (42.9) | 0.79 | 59.8 (36.3) | 57.0 (37.0) | | 57.1 (29.3) | 56.2 (32.8) | 37.8 (23.6) | 31.9 (19.8) | 50.6 (31.6) | 64.2 (55.6) | <0.001 | <0.001 |
| AST, IU/L (±SD) | 36.7 (18.3) | 36.6 (21.0) | 0.58 | 38.9 (19.8) | 36.6 (19.0) | | 38.4 (19.5) | 35.8 (16.4) | 27.7 (14.7) | 25.0 (11.0) | 37.0 (15.5) | 41.9 (27.4) | <0.001 | <0.001 |
| HbA1c, % (±SD) | 6.2 (1.2) | 6.2 (1.1) | 0.87 | 5.8 (0.8) | 6.0 (1.0) | | – | – | 7.0 (1.2) | 6.8 (1.0) | 6.6 (1.4) | 6.4 (1.4) | <0.001 | <0.001 |
| Triglycerides, mmol/L (±SD) | 2.4 (2.0) | 2.4 (2.0) | 0.80 | 2.2 (1.8) | 2.3 (2.3) | | 2.8 (2.7) | 2.4 (1.6) | 2.7 (1.6) | 2.9 (3.2) | 2.2 (1.4) | 2.6 (1.8) | 0.23 | 0.29 |
| Ferritin, ug (±SD) | 50.6 (23.4) | 57.2 (22.7) | 0.24 | 48.7 (25.3) | 59.0 (24.9) | | 50.4 (18.8) | 55.7 (22.2) | – | – | 52.7 (25.7) | 56.4 (20.6) | 0.84 | 0.73 |
| *Imaging markers* | | | | | | | | | | | | | | |
| cT1, ms (±SD) | 829 (106) | | | 805 (112) | | | 845 (98) | | 803 (104) | | 848 (104) | | | |
| PDFF, % (±SD) | 12.4 (8.7) | | | 13.3 (9.3) | | | 10.9 (8.4) | | 10.5 (8.1) | | 13.5 (8.6) | | | |
| *Risk stratification* | | | | | | | | | | | | | | |
| High risk, N (%) | 267 (66) | | | 68 (64) | | | 67 (76) | | 38(46) | | 94 (73) | | | |
| Low risk, N (%) | 115 (28) | | | 35 (33) | | | 24 (27) | | 26 (32) | | 30 (23) | | | |

P-values are reported for comparisons between study arms and in each arm between participating sites.

using a random combination of inclusion criteria, and no significant differences in age, sex, weight, BMI biochemical markers or proportion with T2DM were observed between the two arms (Table 1). In the IA, 66% (267/403) of patients had high liver fat (PDFF ≥ 10%) or evidence of disease activity (cT1 ≥ 800 ms) and were therefore classified as being at high risk of having MASH (Table 1). In this study, 5% (21/403) of patients in the IA had missing cT1 values, due to either the presence of imaging artefacts or high liver fat outside the cT1 quantifiable range.

### Clinical attendances and investigations

Participants had consultations with four types of healthcare specialists: general practitioner, specialist outside/at hospital, and therapist. Patients in the SoC arm (3479) attended significantly more (9%) specialist appointments (with healthcare professionals and for patient assessments) compared to those in the IA arm (3839, p = 0.015) (Table 2). More specifically, patients in SoC had a higher number of consultation visits to specialists including those outside the hospital (248 vs 231), general practitioners (768 vs 711) and therapists (864 vs 559 respectively, p < 0.001) compared to the IA (Table 2). Additionally, there were significantly fewer ultrasound procedures (including VCTE and acoustic radiation force impulse [ARFI]) in the IA (208 procedures) compared to the SoC arm (294 procedures; p < 0.001; Table 2).

Individual unit costs across each country are shown in Table 3. Overall, consultation visits with health care practitioners contributed to the majority of the costs in both SoC (€270,658.62) and the IA (€272,941.15), while the patient assessments accounted for the least (€60,735.23 in SoC vs €52,822.70) (Table 2). In the IA, the additional costs due to the inclusion of MRI (€97,057.07) and mpMRI (€101,148.06) to SoC costs made up 38% of the overall expenditure. Therefore, the total mean expenditures were higher in the IA (€523,968.98) compared to the SoC (€331,393.85) and translated to mean per patient costs of €1,300.17 in the IA and €830.56 in SoC (Supplementary Fig 1). Moreover, those with high risk of MASH according to mpMRI classification incurred higher healthcare resource use expenditure (€1294.27) compared to low-risk participants (€1275.75; p > 0.05).

At the end of the trial, 53% (423/802) of the cohort had a final diagnosis (Fig. 1). When comparing the two trial arms, despite having fewer patient appointments, the IA (57%; 230/403) had significantly higher %-with-diagnosis compared to SoC (48%; 193/399, p = 0.0012; Fig. 2, Table 2).

### Biopsy avoidance

There was no significant difference between the centres in proportion of patients with suspected MASLD incurring liver related hospital consultations and/or liver biopsies (p > 0.05). The referral of patients for biopsy was low, Ulm had the highest number of biopsy procedures in SoC (8 procedures), Leiden and UK had similar rates (5 and 4 procedures respectively) and Coimbra had no procedures (Table 2).

In the IA, 21 (5%, 21/403) patients received a liver biopsy, but 2 had missing data and 1 had comorbid autoimmune hepatitis. As autoimmune hepatitis is an exclusion criterion for MASLD/MASH diagnosis, this patient was excluded from analyses. Thus, of the 18 patients with suspected MASH, following SoC and fibrosis testing, 1 patient was classified as low risk (MAS: 2, fibrosis stage: 0, cT1: 795 ms, PDFF: 5.2%) while the remaining 17 were classified as high risk (either PDFF ≥ 10% or cT1 ≥ 800 ms). Biopsy results showed that 8 patients (40%; 8/20) had MASH with fibrosis, while 10 patients (50%, 10/20) did not meet the histological criteria for this diagnosis and thus could have avoided having a biopsy (Fig. 2). Of 10 patients who did not meet the histological criteria of having MASH with fibrosis, 9 had a cT1 ≤ 875 ms. Hence, 9 (45%, 9/20) biopsies performed in this patient group, indicated as necessary following fibrosis testing and clinician discretion, could have been avoided using mpMRI. The cT1 threshold of 875 ms had a sensitivity of 0.9, specificity of 0.63, positive predictive value of 0.75 and negative predictive value of 0.83 to identify those without MASH with fibrosis who should have avoided having a liver biopsy for MASH diagnosis.

Of those recruited into the SoC arm, 23 patients (6%, 23/399) had a liver biopsy. Of the 23 biopsies performed 11 (48%, 11/23) patients did not have MASH with fibrosis following fibrosis testing.

### Differences in patient care

Notable heterogeneity between sites was observed in patient management and follow-up during analysis. Patients recruited from Portugal (61.2 years) had a higher mean age compared to those from the other centres (Germany: 50 years, Netherlands: 51 years and UK: 53 years; Table 1). Portuguese participants were recruited from a diabetes general practitioner specialist clinic, therefore, 65% of patients had insulin resistance/type 2 diabetes mellitus (T2DM), compared to 28%, 23% and 54% from the Germany, Netherlands and the UK respectively. Although not significantly different (p > 0.05), patients recruited in the UK had higher body mass index (BMI) of 32.5 kg/m² comparable to those from the other 3 sites (~30.4 kg/m²). Additionally, in the IA Leiden and UK had the highest proportion (71.5% and 71% respectively) of participants classified as high risk, whilst Portugal had the lowest (40.5%; Table 1).

The health care resource use response rate was comparable between trial arms. In terms of specialist referral, a reduction in the proportion of patients accessing specialist care (with healthcare professionals and for patient assessments) was seen in the IA in comparison to SoC, in Germany (19%; IA:1090 visit vs SoC:1344 visits) and UK (14%; IA:421 visit vs SoC:487 visits). Notably, patients in Portugal were managed by their general practitioner and relatively few patients (14.0% (11/79) and 9.0% (7/75) for IA and SoC respectively) accessed liver-related specialist care (Table 2; Supplementary Fig 1). Although most patients in UK, Netherlands and Germany were managed by hepatologists, there were no significant differences in the frequency of these visits between the randomisation arms.

### Differences in and rate of diagnosis

In this study, as shown in Fig. 1, patients were diagnosed with either MASLD, MASH or MASH with fibrosis, MASL/MASH with mixed aetiology or other aetiology, therefore, a diagnosis was defined as having a confirmed record of this by the end of the follow-up period. Overall, there was a higher %-with-diagnosis in the IA (57%; 230/403) compared to the SoC arm (48%; 193/399) and was significant in Germany and Netherlands (both p < 0.001, Fig. 2). Conversely, in Portugal where patients were largely managed in primary care only 4% (3/79) of patients in the IA and 3% (2/75) in SoC had a confirmed diagnosis at the end of the follow-up period. In the UK, %-with-diagnosis was comparable in both the imaging and SoC arms (40% (51/127) and 44% (56/128) respectively; Table 2). It is worth noting that irrespective of location, all patients with a final diagnosis after 12-months had weight reduction by incorporation of more physical exercises as part of their treatment.

### Differences in costs associated with liver related consultations and patient assessments

In each country the mean (overall) patient expenditure was greater in the IA (Table 2, Supplementary Fig 2) due to the inclusion of the MRI cost tariffs and mpMRI costs above SoC costs as opposed to any additional health care resource use. For instance, when comparing the overall expenditure, general practitioner, specialist, and patient assessment costs were all significantly lower in the IA compared to the SoC arm (Table 2). Moreover, in Germany, Netherlands and UK, a reduction in the use of both ultrasound and transient elastography was observed in the IA compared to SoC (Table 2). In Germany and Netherlands, additional expenditure resulted from a slightly greater number of specialists at hospital consultations for those classified as high risk. Such consultations may be deemed as high value care if associated with a high-risk patient warranting further specialist evaluation. When overall patient costs were compared between high and low risk groups, the costs were similar between the two groups with those in Netherlands and UK being numerically higher in the high-risk group (Table 2; Fig. 2; Supplementary Fig 2).

**Table 2 | Aggregate (total and regional) number of visits with health care practitioners (general practitioner, specialist outside hospital visit, specialist at hospital visit and therapist) and patient assessments (blood tests, biopsy and ultrasound testing) between the imaging arm (SoC + mpMRI) and standard of care arms**

| | Cohort a (€) | | Ulm, Germany (€) | | Leiden, The Netherlands (€) | | Coimbra, Portugal (€) | | UK (€) | |
|---|---|---|---|---|---|---|---|---|---|---|
| | Imaging | SoC | Imaging | SoC | Imaging | SoC | Imaging | SoC | Imaging | SoC |
| **Number of patients, N** | 403 | 399 | 107 | 109 | 89 | 88 | 79 | 75 | 128 | 127 |
| **Visits with health care practitioners** | | | | | | | | | | |
| *General Practitioner* | | | | | | | | | | |
| Total consultations | 711 | 768 | 158 | 209 | 233 | 212 | 171 | 174 | 149 | 173 |
| Total costs | 33,441.70 | 39,308.60 | 9359.20 | 12,378.60 | 3645 | 3311 | 769.50 | 783.00 | 19,668 | 22,836 |
| *Specialist outside hospital* | | | | | | | | | | |
| Total consultations | 231 | 248 | 124 | 172 | 85 | 41 | 10 | 12 | 12 | 23 |
| Total costs | 40,362.70 | 37,140.50 | 13,944.70 | 19,329.40 | 22,761 | 11,024 | 309 | 370.10 | 3348 | 6417 |
| *Specialist at hospital* | | | | | | | | | | |
| Total consultations*** | 857 | 769 | 317 | 253 | 373 | 333 | 51 | 73 | 116 (0.95) | 110 (0.93) |
| Total costs | 169,647.70 | 150,834.50 | 35,595.30 | 28,443.50 | 100,114 | 89,447 | 1574.4 | 2254 | 32,364 | 30,690 |
| *Therapist* | | | | | | | | | | |
| Total consultations*** | 559 | 864 | 183 | 307 | 286 | 498 | 64 | 3 | 26 (0.21) | 56 (0.47) |
| Total costs | 20,045.20 | 32,943.70 | 10,872.20 | 18,213.7 | 6979 | 12,162 | 1024.0 | 48.0 | 1170 | 2520 |
| **Patient assessment** | | | | | | | | | | |
| *Biopsy* | | | | | | | | | | |
| Total procedures | 21 | 23 | 9 | 8 | 11 | 5 | 0 | 0 | 1 | 4 |
| Total costs | 17,538 | 7582 | 1168 | 1044 | 8062 | 3634 | 0 | 0 | 726 | 2904 |
| *Bloods* | | | | | | | | | | |
| Total tests | 892 | 873 | 235 | 266 | 354 | 261 | 85 | 112 | 218 | 234 |
| Total costs | 25,521 | 28,248 | 22,159 | 25,111 | 1762 | 1298 | 510 | 669 | 1090 | 1170 |
| *Ultrasound* | | | | | | | | | | |
| Total procedures*** | 208 | 294 | 64 | 129 | 86 | 90 | 17 | 22 | 41 | 53 |
| Total costs | 17,043.07 | 24,224.87 | 5901 | 12,028.70 | 8,899.60 | 9,304.30 | 353.10 | 433 | 1619 | 2107 |
| **Patients with a final diagnosis** | | | | | | | | | | |
| Patients with a diagnosis, N ** | 230 | 193 | 87 | 69 | 84 | 71 | 3 | 2 | 56 | 51 |
| **Total expenditure** | | | | | | | | | | |
| Visits with health care practitioners | 272,941.15 | 270,658.62 | 69,771.40 | 78,365.20 | 133,499.00 | 115,944.00 | 3,676.90 | 3,455.10 | 56,550.00 | 62,463.00 |
| Patient assessment | 52,822.70 | 60,735.23 | 29,228.00 | 38,183.70 | 18,722.95 | 14,236.30 | 863.10 | 1102.00 | 3,435.00 | 6181.00 |
| MRI | 97,057.07 | | 36,219.00 | | 35,798.00 | | 7290.00 | | 15,210.00 | |
| mpMRI | 101,148.06 | | 23,177.00 | | 25,284.00 | | 17,460.00 | | 30,186.00 | |
| Total cost | 523,968.98 | 331,393.85 | 158,395.40 | 116,548.90 | 213,303.95 | 130,180.30 | 29,290.00 | 4557.10 | 105,381.00 | 68,644.00 |
| Per patient*** | 1300.17 | 830.56 | 1480.33 | 1069.26 | 2379.87 | 1479.32 | 370.76 | 60.76 | 823.29 | 540.50 |
| Per high-risk patient | 1294.27 | | 1483.39 | | 2437.76 | | 353.20 | | 822.20 | |
| Per low-risk | 1275.75 | | 1493.44 | | 2399.67 | | 360.49 | | 807.92 | |

a £ changed to € using the 2020 exchange rate for total cohort expenditure costs.
Cumulative costs incurred in both trial arms showing total aggregate health care resource utilisation are also summarised. All significantly different comparisons between the imaging and SoC arms are denoted with the following levels of significance: *p < 0.05; **p < 0.01; ***p < 0.001.

**Table 3 | Unit Health Care costs (Nominal value)**

| Site | GP | Therapist | Specialist | Blood | Ultrasound | MRI | Elastography | Biopsy | mpMRI |
|---|---|---|---|---|---|---|---|---|---|
| Coimbra (Portugal) €[21] | 4.5 | 16 | 31 | 6.71 | 19.52 | 127.9 | 25.7 | 285.6 | 301 |
| Ulm (Germany) €[22] | 59.45 | 59.45 | 112.84 | 94.77 | 92.67 | 470.38 | 100 | 131.1 | 301 |
| Leiden (The Netherlands) € | 15.7 | 24.5 | 269.81 | 5.0 | 105.51 | 425.8 | 100 | 732.85 | 301 |
| UK £[23] | 132 | 45 | 279 | 5.0 | 39 | 130 | 49 | 726 | 258 |

Unit prices for Portugal (Ministerio de saude Portugal and Portugal Health Care Pricing (NHS)), Germany (Institut für das Entgeltsystem im Krankenhaus), and UK (the NHS national tariff payment system) were obtained from publicly available resources/tarrifs. Unit prices for the Netherlands were obtained from the Leiden University Medical Center hospital pricing/cost of care.

## Cost-effectiveness of associated with the use of mpMRI in adults with suspected MASLD

Considering the total costs for the IA (€523,968.98) compared to the SoC (€331,393.85) and the difference in the proportion of patients with a final diagnosis (%-with-diagnosis) (57% and 48% respectively), the ICER across the cohort was €5067.77. Using a QALY loss of 0.03 per patient per missed diagnosis[30], the use of mpMRI in the IA was cost-effective with an ICER of €5067.77 per 1.02 QALYs gained (€4968.40/QALY gained). Further analyses to evaluate the minimum and maximum cost effectiveness range showed that for a conservative ICER of ≤£10,000 per QALY gained, the inclusion of mpMRI was cost effective between €301 – €1204. Using the more widely reported ICER of ≤~£20,000 per QALY gained thresholds reported in literature[29–31], a cost range of €301-€2709 for mpMRI was within the cost effectiveness ratio.

## Discussion

In this prospective, randomised controlled, multi-national European trial we investigated the impact on patient management including mpMRI in the MASLD clinical pathway would have in a real-world setting. We identified four key findings. Firstly, regarding the primary objective of this study, in a population with suspected MASLD, mpMRI is a cost-effective tool with an ICER of €4929/QALY gained. Secondly, with regards to liver-related consultations, when using mpMRI as part of patient management, there are significantly less clinical consultations and follow-up appointments with healthcare specialists when compared to standard of care alone. Additionally, the proportion of patients with a final diagnosis by the end of the trial follow-up period was significantly higher in the imaging arm compared to SoC alone implying that mpMRI has a significant impact on rate of diagnosis. Third, mpMRI identified 50% of patients who did not require a biopsy for the diagnosis of MASH thereby showing clinical utility to support patient stratification for those with suspected MASLD. Lastly, there is an opportunity to optimise the clinical efficiency and long-term costs of MASLD clinical care as notable differences in real-world patient care highlight the need for streamlined management.

Liver biopsy is the reference standard diagnostic procedure for MASH, nevertheless, it is no longer widely used in routine clinical care and is no longer required for the diagnosis of MASLD[32]. In this trial, although no significant differences were observed in the overall liver biopsy rates, Ulm was found to have the highest number of biopsy procedures whilst Coimbra had no patients biopsied. Differences in management have been noted in literature with <25% of clinicians requiring liver biopsy to make a MASH diagnosis[33]. Moreover, literature shows there is variance in patient management not only between centres, but also with recommended clinical guidance[33–35]. In this trial, only 6% of patients had a referral for a liver biopsy. Juxtaposing this reduction in the proportion of patients being biopsied with the increase in MASLD disease prevalence highlights the clear need to establish the role of alternative noninvasive techniques (NITs) to support patient diagnosis and management. NITs have potential to substantially impact the streamlining of MASLD patient management pathways outside of clinical trial settings.

Current tools (MRE, VCTE, SWE and ultrasound) are good at identifying advanced fibrosis but not at detecting MASH; a need highlighted in

this trial as over half of the patients referred for a biopsy did not have high-risk disease (MASH with fibrosis) and thus could have avoided the invasive procedure. However, although fibrosis detection makes up the basis of the majority of risk predictors, MASH is equally as likely to progress to an adverse clinical event[13,15]. cT1 has shown clinical utility to support MASH diagnosis by enabling better risk assessment of high-risk MASH (at-risk MASH)[17,36], identifying those most likely to experience adverse outcomes[15,26] and in monitoring treatment responses in those with MASH[8,17,37] in therapies with different mechanisms of action. Regulatory bodies such as the FDA recognise the need for NITs to support patient management and have introduced the biomarker qualification program (BQP) to support adoption of NITs. Although there are no currently approved pharmacotherapies for MASH, the need for companion diagnostics to support management is still recognised. Currently, cT1 is the first biomarker to have an accepted qualification plan in the FDA BQP for approval as a diagnostic enrichment biomarker in MASH clinical trials[38]. Moreover, to this end, the clinical utility of NITs (individually, in sequence or in combination) for differing contexts of use is being heavily investigated by various consortia such as LITMUS, NIMBLE, and NAIL-NIT.

Considering costs, when all patient assessments were compared, SoC (€331,393.85; €830.56 per patient) incurred overall lower costs compared to the imaging arm (€523,968.98; €1,300.17 per patient). However, despite having a higher cost in the short-term, the use of mpMRI provided additional actionable clinical information to support patient clinical management which resulted in significantly fewer specialist consultations, fewer additional patient assessments, and higher proportion of patients with a final diagnosis (%-with-diagnosis). These findings are in-line with the latest economic assessments of screening strategies for high-risk MASLD which show that screening for high-risk MASLD is cost-effective[39]. Furthermore, although initially cost incurring, the use of NITs for longitudinal assessment of patients reduces long-term costs associated with patient management[39]. It is worth noting that the cost of liver biopsy included herein do not cover associated complications, such as pain (12.9%), minor complication (9.53%) and major complication (2.44%)[3]. These, in addition to associated hospital bed and care costs, pain management costs, as well as patient out-of-pocket (OOP) costs, all have an impact on the cost of healthcare for this ever-growing population. The epidemics of obesity and type 2 diabetes contribute to global prevalence of MASLD. Delayed, missed or under-diagnosis can have a negative impact on disease progression which could lead to adverse clinical outcomes[6] and hospitalisation[3]. To put this into context, significantly more patients received a final diagnosis within 12 months in the imaging arm compared to the SoC arm (57% vs. 48% respectively). It can be argued that patient care is multi-faceted, as no one test can be used to confirm diagnosis, however, these diagnosis rates were at no additional liver-related healthcare resource use. With this, commissioning bodies, payors and insurance providers should consider both the short and long-term costs, resource use (including reduction in specialist appointments), and clinical effectiveness (including the improvements in quality and performance) brought on by the addition of new tests. Clinical practise guidelines should also consider the inclusion and implementation of these tests to support patient management[39]. This is especially necessary as earlier and more accurate diagnosis can facilitate for intervention which is less

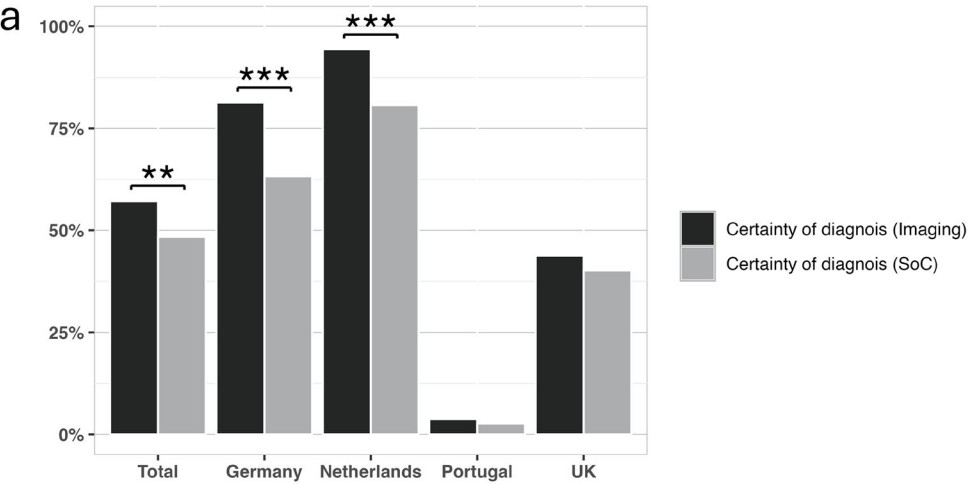

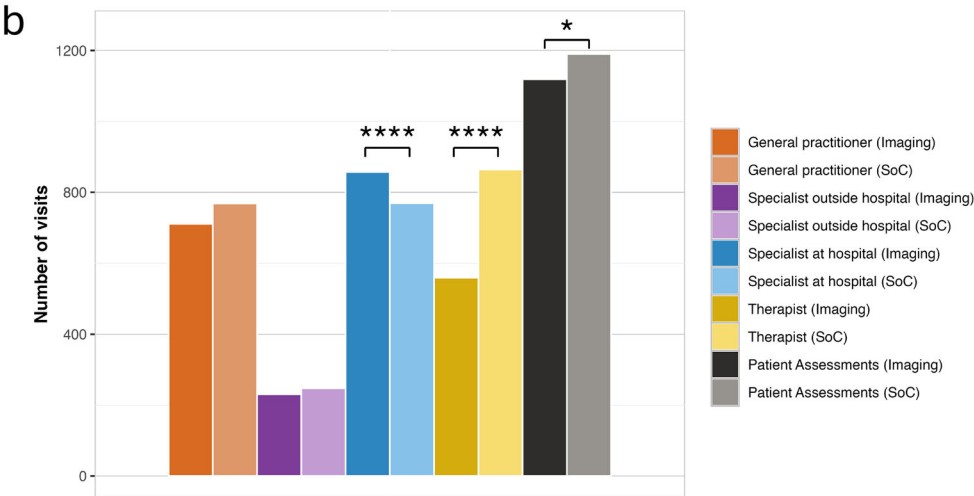

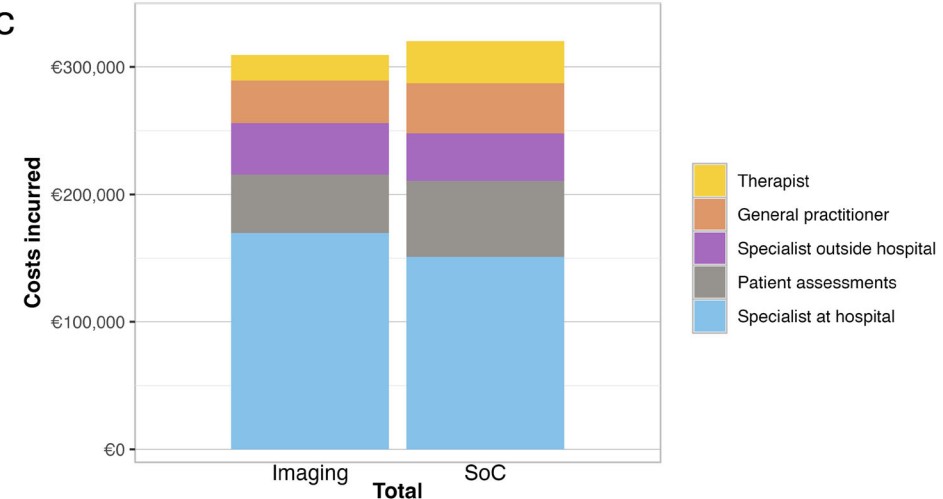

**Fig. 2 | Proportion of patients in the imaging and standard-of-care arms who received a final diagnosis by the end of the trial, along with the total number of healthcare visits, patient assessments, and associated costs in the standard-of-care management for both arms.** a Proportion of patients in the imaging and standard-of-care (SoC) arms (total/whole cohort and regional) who received a certain diagnosis by the end of the trial period. Aggregate (**b**) number of visits with health care practitioners (general practitioner, specialist outside hospital visit, specialist at hospital visit and therapist) and patient assessments (blood tests, biopsy and ultrasound testing) between the imaging arm (SoC + multiparametric MRI) and standard of care arms, and (**c**) costs incurred in the standard of care management of patients in the imaging arm (without inclusion of MRI costs) and the SoC arm. All significantly different comparisons between the imaging and SoC arms are denoted with the following levels of significance: *$p < 0.05$; **$p < 0.01$; ***$p < 0.001$. There were $N = 802$ participants in the whole study, $N = 403$ imaging arm (Germany: 107, Netherlands: 89, Portugal: 79, UK:128), and $N = 399$ in the SoC arm (Germany: 109, Netherlands: 88, Portugal: 75, UK: 127).

costly and has the potential to lower long-term management costs (direct costs estimated to be >€35 billion in Europe) and DALYs (currently at ~174,564).

Sequential testing algorithms, some of which have recently been shown to be cost effective[39], have been proposed to support better patient diagnosis and management of MASLD/MASH[19,32]. For instance, the National Institute for Health and Care Excellence, which provides expert evidence-based best practice for health and care practitioners and whose guidance is targeted at ensuring the best care is delivered to patients in a fast manner whilst ensuring value for the taxpayer, encourages the uptake of best practice to improve outcomes. In their assessment[29,30], NICE considered an ICER of ≤£20,000 per QALY gained as cost effective. Although a conservative approach was pursued in this study, with an ICER of ≤£10,000 per QALY gained considered as cost effective, the results indicate that the addition of mpMRI to the management plan to support disease (MASLD/MASH) diagnosis provides incremental gains of QALYs over SoC. By supporting diagnosis of significantly more patients, with the impending regulatory approval of therapies to treat MASLD/MASH for which cT1 and PDFF have shown utility to assess treatment response[40], early diagnosis of patients has the potential to limit the transition of patients towards more severe clinical outcomes (such as liver cirrhosis and end-stage liver disease) and their associated quality-of-life and economic costs[31].

This study has several strengths and some limitations. Firstly, no additional incidental findings were identified following the addition of the MRI scan. Moreover, although there may be perceived limitations around the introduction of additional NITs to support MASLD patient management, clinical practice guidelines are actively calling for the evaluation of additional tests which can help streamline patient management. The findings from this trial increase the value of real-world evaluations (which are not standardised to a 'virtual' idealistic standard of care) as they highlight the existing differences in current patient care whilst also highlighting the real unmet needs which NITs can support to address. Hence, it is anticipated that MRI tools will be complementary to existing tools in the MASLD pathway and will support patient stratification/triage and biopsy avoidance. For instance, similar to that seen in clinical practise today, there was a low number of biopsies performed in both study arms. Patient management has evolved over the years, and with the high prevalence of MASLD/MASH coupled with the change in diagnostic criteria, liver biopsy is no longer used in all cases to confirm diagnosis[32–36]. The aim of this study included evaluation of the clinical utility of using mpMRI and thus, as all patients underwent SoC assessment, the rate of use of other techniques (including FIB-4, VCTE, ELF, ARFI, etc.) was assumed to be similar in both study arms.

Our findings quantified the drivers of health care resource use for MASLD patients, an aspect which has not been detailed in recent studies that have endeavoured to explore the burden of MASLD[39]. There is no consensus on how to best make use of large samples of economic data when healthcare systems and pricing across jurisdictions vary, and health technology assessment (HTA) decision makers require clinical and economic evidence pertaining to their own nation. However, we endeavoured to show the real-world multi-national health economic cost implications, resource utilisation and proportion of patients with a final diagnosis including mpMRI in the MASLD pathway would bring. Importantly, complications resulting from liver biopsy, and any resultant additional care, were not included in the cost analyses and therefore could have further impacted reported costings. Similarly, when considering the MRI costs, only the costs for scan time was included for assessment, therefore no costs associated costs (including clinician time, radiology time, assessment for incidental findings, and other overheads) were included in the MRI costs presented herein. In relation to collected health care resource use, although recall bias maybe be at play it was assumed to impact groups comparatively. Data missingness can be a source of potential bias as it can reduce the size of the sample available for analysis. Alongside the overall assessment of data missingness, in cases where over 10% of data is missing across multiple

variables, sensitivity analyses are sometimes required to evaluate potential impact on analysis results and account for sources of bias[41]. As this was a randomised control trial with <10% overall missing data, available data was considered to be within the acceptable range of allowable missing data in healthcare investigations and thus no additional sensitivity analyses were required. It is worth noting that in some cases the causes of missing data cannot be addressed as they are themselves unmeasured and multifaceted. This can further be complicated in multi-institutional multi-national studies such as that presented herein. Nevertheless, future studies should make note of these limitations and the potential impact they could have on reported results[42]. Lastly, cost effectiveness assesses the degree to which something is effective or productive in relation to its cost. In this study, we assumed that the healthcare systems from the various countries were similar to the NHS i.e. using tax income to provide "free" essential medical services, and this assumed a QALY loss due to undiagnosed MASH of 0.03. Furthermore, we excluded patients with other comorbid diseases, including autoimmune liver disease, cirrhosis, Wilsons disease, cystic fibrosis related liver disease and viral hepatitis, as these patients will likely follow a different management pathway. Therefore, although these patients will constitute a proportion of those with MASLD/MASH, their management will not likely follow the same pathway due to the presence of other comorbid diseases. Nevertheless, as shown in literature, these patients still benefit from an mpMRI[43–46]. Future work should look at quantifying the QALYs for each study arm allowing for subgroup analysis for each country and to obtain a more accurate QALY loss due to undiagnosed MASH.

## Conclusion

In conclusion, we investigated the clinical utility of adding mpMRI to MASLD standard of care multi-nationally to support patient diagnosis. We evaluated the proportion of patients requiring additional clinical consultations, and the proportion of patients obtaining a diagnosis during the study period between the randomisation arms. Furthermore, we determined the cost impact of using mpMRI compared to current standard of care by examining the differences in the types and magnitude of diagnostic healthcare resource use, the cost of visits with health care practitioners and the proportion of avoidable biopsies incurred (in low-risk patients). Our findings showed that the inclusion of liver mpMRI is cost-effective and can lead to fewer specialist consultations, support patient stratification, avoid biopsy and increase the rate of diagnosis without any additional liver-related health care resource use. Widespread use of non-invasive tools can support optimisation of the MASLD clinical pathway and improve patient management by providing clinically effective healthcare in patients with suspected MASLD.

## Data availability

The data and analytic methods used in this study remain the property of the study sponsors. All de-identified participant data may be made available to other researchers upon request following permission, investigator support and following a signed data access agreement. Source data underlying Fig. 2a, b, and c can be found in Table 2.

## Abbreviations

| | |
|---|---|
| ARFI | Acoustic Radiation Force Impulse |
| BMI | Body Mass Index |
| CPI | Consumer Price Index |
| cT1 | corrected T1 |
| EASL | The European Association for the Study of the Liver |
| EU | European Union |
| F0-4 | Fibrosis levels 0-4 |
| HCRUQs | Health Care Resource Use Questionnaires |
| HTA | Health Technology Assessment |
| ICER | incremental cost-effectiveness ratio |
| mpMRI | Multiparametric Magnetic Resonance Imaging |
| MRE | Magnetic Resonance Elastography |
| MRI | Magnetic Resonance Imaging |

| MASLD | metabolic dysfunction–associated liver disease (MASLD) |
| MASH | metabolic dysfunction–associated steatohepatitis (MASH) |
| MAS | metabolic dysfunction–associated liver disease activity score |
| NHS | National Health Service |
| OECD | Organisation for Economic Co-operation and Development |
| ONS | Office for National Statistics |
| PDFF | Proton Density Fat Fraction |
| QALY | quality-adjusted life year |
| RADIcAL-1 | Non-invasive rapid assessment of non-alcoholic fatty liver disease using magnetic resonance imaging with Liver*MultiScan* |
| SoC | Standard of Care |
| SWE | Sheer Wave Elastography |
| T1 | longitudinal relaxation time |
| T2DM | Type 2 Diabetes Mellitus |
| UK | United Kingdom |
| VCTE | Vibration Controlled Transient Elastography |

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

## Acknowledgements

The authors wish to thank the participants who took part in the research. CDB was supported in part by the Southampton National Institute for Health Research Biomedical Research Centre (IS-BRC-20004), U.K. This paper presents independent research titled Non-invasive Rapid Assessment of NAFLD Using Magnetic Resonance Imaging with LiverMultiScan (RADIcAL1) supported by a European Union's Horizon 2020 SME Instrument Phase 2 Program under grant agreement No 719445.

## Author contributions

Funding acquisition: M.M.D., M.J.C., H.J.L., R.B. Recruitment, data collection and data entry oversight: M.H.M., C.D.B., J.P., S.A., K.A., J.B., M.J.C., J.J.S., A.F., M.C.B., S.B., M.M.D., D.J.C., D.F., H.J.L. Data quality reports: MF, ES. Data verification: J.E.C., E.S., M.F., H.J.L. Data analysis: J.E.C., E.S., M.F., C.B. Statistical Methodology: J.E.C., P.L. Manuscript drafting: J.E.C., E.S., M.F. Manuscript review and editing: J.E.C., E.S., M.F., C.B., P.L., D.T., R.B., M.H.M., C.D.B., J.P., S.A., K.A., J.B., M.J.C., J.J.S., A.F., M.C.B., S.B., M.M.D., D.J.C., D.F., H.J.L. All authors reviewed, discussed, and agreed with the manuscript.

## Competing interests

The authors declare the following competing interests: MF, ES, CB, and RB are employees of Perspectum Ltd. DT is a consultant for Perspectum Ltd. JEC is a student at University College London doing a Knowledge Transfer Program (KTP) with Perspectum Ltd. All other co-authors declare no competing interests relevant to this work. The funder did not have a role in the study design, data analysis or manuscript preparation. The views expressed, are those of the author(s) and not necessarily those of the funding body (European Union's Horizon 2020).

## Additional information

Elizabeth Shumbayawonda ®[1] ✉, Marika French[1], Jane Elizabeth Carolan[1,2,3], Cayden Beyer[1], Paula Lorgelly[2], Dimitar Tonev[1], Rajarshi Banerjee[1], Michael H. Miller[4], Christopher D. Byrne ®[5,6], Janisha Patel[5,6], Saima Ajaz[7], Kosh Agarwal ®[7], Johanna Backhus[8], Minneke J. Coenraad[9], Jelte J. Schaapman ®[9], Andrew Fraser[10], Miguel Castelo Branco ®[11], Stephen Barclay[12], Matthias M. Dollinger[8], Daniel J. Cuthbertson[13], Daniel Forton[14] & Hildo J. Lamb ®[15]

[1]Perspectum Ltd, Oxford, UK. [2]Institute of Health Informatics, University College London, London, UK. [3]Institute of Epidemiology and Health Care, University College London, London, UK. [4]Ninewells Hospital, Dundee, UK. [5]Southampton National Institute for Health and Care Research Biomedical Research Centre, University Hospital Southampton, Southampton General Hospital, Southampton, UK. [6]Nutrition and Metabolism, Faculty of Medicine, University of Southampton, Southampton, UK. [7]Institute of Liver Studies, Kings College Hospital, London, UK. [8]University Hospital Ulm, Ulm, Germany. [9]Department of Gastroenterology and Hepatology, Leiden

University Medical Center, Leiden, The Netherlands. [10]Consultant Hepatologist and Gastroenterologist, Queen Elizabeth University Hospital, Glasgow, UK. [11]CIBIT (Coimbra Institute for Biomedical Imaging and Translational) Research, Faculdade de Medicina, Instituto de Ciências Nucleares Aplicadas à Saúde, Universidade de Coimbra, Coimbra, Portugal. [12]Glasgow Royal Infirmary, Glasgow, UK. [13]Department of Cardiovascular and Metabolic Medicine, University of Liverpool, Liverpool, UK. [14]Department of Gastroenterology and Hepatology, St. George's Hospital, London, UK. [15]Department of Radiology, Leiden University Medical Center, Leiden, The Netherlands. ✉e-mail: elizabeth.shumbayawonda@perspectum.com

