## [Peer review file · Communications Medicine]

Multiparametric MRI increases rate of MASLD diagnosis with fewer specialist appointments and interventions: a real-world multi-national randomised clinical trial

Corresponding Author: Dr Elizabeth Shumbayawonda

Version 0:

Reviewer comments:

Reviewer #1

(Remarks to the Author)

Thank you for the opportunity to review this paper.

The introduction is fantastic and really lays the groundwork very well. The only comment here is about cT1. The introduction would benefit from some additional lead in regarding this characteristic (how its calculated/what it is/how mpMRI allows you to calculate cT1). It may be a very simple addition but since it is a modality of importance to this paper, a bit more background would be nice.

For what its worth, the real world nature is refreshing.

Page 9, line 316, what does AI refer to? This paragraph is written a little wordy and hard to follow. Recommend reframing the results by study arm.

Section 3.5: diagnosis of what? NAFLD, NASH, NASH with fibrosis? Also with nomenclature change, this would be a good time to mention that you may want to change to MASLD, MASH terminology.

Page 11, line 384: "clinician the proportion"? something is incorrect here.

Why did IA had less consultations/follow up with specialists than SoC? Is the idea that mpMRI should be used by PCPs/general practitioners but then NOT refer to hepatology when there is MASH +/- fibrosis. This doesn't add up. Furthermore, not every recruited patient came from a primary care pathway, so how is it that they did not see more specialists/clinical consultations? This part is a little unclear to me.

Many believe MASH is a biopsy proven diagnosis. Thus understanding how cT1 makes the diagnosis of MASH is extremely important here. Somehow this was only mentioned in the introduction.

Reviewer #2

(Remarks to the Author)

Shumbayawonda et al. describe a study investigating the cost-effectiveness of performing multiparametric MRI to evaluate patients with suspected non-alcoholic fatty liver disease (recently renamed to metabolic dysfunction associated steatotic liver disease, MASLD). A large, multi-centre, multi-national trial was conducted for this, including centres from Portugal, Germany, the Netherlands and the UK, including 802 patients. The authors found that compared to the standard-of-care patient pathway, patients undergoing mpMRI had fewer specialist appointments and patient assessments, as well as a higher proportion of them getting a definite diagnosis within a year. Whilst the short-term per-patient cost of including mpMRI in the patient pathway was higher than for standard-of-care, the authors argue that this difference will pay off in long term patient treatment/monitoring/management costs.

I only have a few small comments for the authors:

1. Please change NAFLD/NASH to MASLD/MASH throughout the paper to conform with the new nomenclature.
2. Histology is used to stage fibrosis, not to score it. Please replace references to "fibrosis score" with "fibrosis stage" throughout the manuscript.

3. Please include patient numbers referred for specialist appointments as well as patients with a diagnosis in the two groups - if this doesn't fit within the word limit of the manuscript, report them at least in the supplement
4. P4L144: I suggest changing "≥ 3" to "3 or more" to help clarity
5. Please include sequence parameter details of the mpMRI protocol or refer to a previous study describing these. A more detailed description of the image analysis part is also required.
6. Were the MR imaging read by radiologists for incidental findings or was mpMRI the only exam performed?
7. I suggest that the authors add the number of patients in the header of Table 2.

Reviewer #3

(Remarks to the Author)

The manuscript by Shumbayawonda et al, 'Multiparametric MRI increases rate of NAFLD diagnosis with fewer specialist appointments and interventions: a real-world multi-national randomised trial' presents an analysis of the RADICAL-1 clinical trial, concluding that sole reliance on multi-parametric MRI in comparison to 'Standard of Care' would reduce cost, specialist consultations, and unnecessary biopsies.

It is laudable and no small feat to carry out a multi-national clinical trial across 10 different sites. However, there are a few points, that if clarified, could help with the clarity and confidence of the reported conclusions.

As a point of clarification, obtaining an MRI is considered standard of care in workup of patients suspected NAFLD and the studies are typically done with some type of post-contrast T1 sequence as well as a sequence to assess fat content (PDFF in this trial). I could not find it explicitly stated, however does the SoC MRI not include these two sequences? Is the purpose of this study then to assess inclusion of cT1 and PDFF as part of the imaging workup?

Table 2 lists the total cost of MRI as €97,057.07 with mpMRI as an additional €101,148.06. Does this mean that the addition of those two sequences more than doubled the MRI cost?

How was the gold standard for NAFLD defined? Histopathology is typically the definitive means. Line 241 of the manuscript has the first use of the NAS abbreviation -- this presumably refers to the NAFLD Activity Score -- however it is never defined (should be defined in text and added to abbreviation list). It is not clear how the NAS could be defined if biopsies were never obtained in the vast majority of cases.

What was the initial target enrollment for the study?

Statistical analysis states "between group", however cost differences between different countries is significant. It seems that it may have been more appropriate for comparison between the two arms for on a country by country basis. When looking at these differences, there are very different trends for each country, likely attributable to different implementations of health care models.

The protocol states that 1.5 or 3T scanners were used; there can be significant differences for cT1 and PDFF obtained from these. How was variability for these differences accounted for?

I don't follow the logic for how an imaging measurement with a PPV of 0.75 and NPV of 0.83 could have supplanted any histopathological characterization in such a small sample size.

Minor grammatical/typographical errors:

Line 384

Table 1: T2DB presumably should read T2DM

Reviewer #4

(Remarks to the Author)

The authors should be commended for delivering a pragmatic real-world multinational randomised trial of multiparametric MRI in the evaluation of patients with NAFLD compared to standard of care (which was institution specific and not defined) in challenging circumstances with follow-up being curtailed because of the pandemic.

However, there are a number of significant methodological concerns in the paper. Different health economic evaluations are mentioned and appear to be used interchangeably despite being radically different things. The abstract starts off by stating that the study team investigated the utility and cost of mpMRI (utility is a very specific definition in health economics and is not described any further in the body of the article). The primary objective of the study was to measure the cost-effectiveness of mpMRI but no cost-effectiveness analysis is presented. At best, a cost-minimisation analysis is described (though not explicitly defined as such and is not clearly articulated).

Another major limitation is that the field of NITs has moved dramatically since the timeline described in the trial. In the current era of expanded step-wise use of NITs, any potential cost saving with mpMRI in the field of MASLD is likely to be extremely limited or non-existent.

Reviewer #5

(Remarks to the Author)

In the study titled “Multiparametric MRI increases rate of NAFLD diagnosis with fewer specialist appointments and interventions: a real-world multi-national randomised trial” Shumbayawonda Et Al. conducted a randomized trial with the primary objective of evaluating the utility and cost of using mpMRI to risk stratify adults with suspected NAFLD in a multinational setting. The paper is well written. However, the methods and results had multiple major problems that potentially could impact the overall conclusion.

Introduction

- Authors should briefly recognize the NAFLD/NASH new nomenclature in their disease definition.
- Lines 107 to 109, the authors stated the following: “Studies have demonstrated the improved ability of cT1 in identifying and stratifying disease when compared to other noninvasive technologies”¹. The study cited was done in a Japanese cohort, which should be stated. In addition, the authors need to expand on what they mean by “disease” and which noninvasive technologies were investigated.
- o 1. Imajo, K., Tetlow, L., Dennis, A., Shumayawonda, E., Mouchti, S., Kendall, T.J., et al. (2021) ‘Quantitative multiparametric magnetic resonance imaging can aid non-alcoholic steatohepatitis diagnosis in a Japanese cohort’, World Journal of Gastroenterology, 27(7), pp.609-623. doi:10.3748/wjg.v27.i7.609

Methods

- In line 148, the authors stated the following in the exclusion criteria “alcohol over-use/abuse and any other cause of disease”. What disease are they referring to?
- Power analysis needs to be included as part of the statistical analysis section
- Case wise deletion assumes Missing Completely at Random (MCAR), which is unlikely to hold in this study. Proportions missing and their predictors need to be investigated, and imputation needs to be done as a sensitivity analysis. Otherwise, more justifications need to be provided for the use of a case-wise deletion strategy.
- Given the missing patterns in the data and differences in SoC access sites, it would be more appropriate to examine the rate of use per patient. In addition, all outcome analyses should adjust for sites.

Results

- Study design was 1:1; however, the SoC group had 4 patients less than the intervention arm. An explanation for this needs to be included in the results section 3.1. More importantly, a full description of the final analysis sample needs to be included here. According to the
- To assess overall balance and potential difference in patient characteristics between participating sites, Table One should include p-values to support the claim of no statistically significant differences between 1) intervention and SoC groups and 2) overall difference between participating sites. Variables that are significantly different need to be accounted for in the outcome models.
- Tables must include the number of patients included in the analyses.
- Using simple arithmetic to determine the number of patients used for analyses in Table Two, it seems that the intervention arm had 345 patients while the SoC group had 365. If these numbers are correct, then using aggregate numbers is inaccurate. Instead of reporting total numbers, the rate of consultations should be estimated using poisson regression while accounting for the size of each site.
- It also seems that table one does not describe the analytic sample used in table 2 given the differences in the numbers of participants in each group. Table Two was done on 710 participants, and table one describes 802. This is problematic in an intention-to-treat analysis.
- General comment: Tables and figures titles should be limited to titles with no result interpretations. For example, the Table One title should be limited to “Table 1 – Overall and regional baseline demographic summary.” The following should be part of the results “Despite variations in patient management, there were no significant differences in age, sex, weight, BMI, biochemical markers or proportion with T2DM observed between the two trial arms.”

Version 1:

Reviewer comments:

Reviewer #1

(Remarks to the Author)

My comments have been addressed

Reviewer #2

(Remarks to the Author)

I would like to thank the authors for amending their manuscript, which now reads really well. I only have two small comments:

1. Since the mpMRI protocol was used on all three major MRI vendors, I suggest the authors remove the reference to SENSE in Table S3, as GRAPPA is likely the acceleration method of choice on Siemens machines.

2. Please correct the reference errors in Table 3.

Reviewer #3

(Remarks to the Author)

The authors have adequately addressed the concerns.

Reviewer #4

(Remarks to the Author)

Thank you for your response to my review comments. I am satisfied that these have been appropriately addressed.

Version 3:

Reviewer comments:

Reviewer #6

(Remarks to the Author)

In general this seems like an important and well conducted study. However, the results might be presented more carefully. The paper should clearly report the primary outcome and all secondary outcomes of the trial, indicated as such. These primary and secondary comparisons should be of the as-randomized study population, without excluding any patients (such as those who did not get a high quality image or biopsy-they still count!).

Additional outcomes beyond the primary and secondary outcomes of the trial should be clearly indicated as post hoc and exploratory, and should not be reported in the abstract.

For example, the proportion of patients "incurring liver-related hospital consultations or liver biopsies" should include all patients, even those in the imaging arm without successful imaging. (This was the primary outcome). Similarly, the proportion of patients with a final diagnosis should include those patients who had a failed imaging study or failed biopsy.

Please double check that statements in the abstract are well supported by the data- this did not always seem to be the case.

Please limit statements in the conclusion to primary or secondary trial outcomes.

Specific comments.

Abstract.

1. Please give effect sizes with 95% confidence intervals (e.g. percentage point differences in rates) rather than p values.
2. Please report the primary outcome of the trial in the abstract and report all secondary outcomes. The primary outcome was not different between arms
3. Across all involved hospitals, %diagnosis was significantly higher in the imaging arm ($p=0.0012$). I did not find that this statement was supported by the results (fig 2). The underlying data should be reported, and this analysis should use all patients, as randomized.

Conclusion:

4. Please rephrase the conclusion in terms of the primary outcome of the trial and the cost effectiveness.
5. Please rephrase "without any additional health care resource use" as "without increase in other liver-related health care resource use"
6. Please include the increase in cost in the conclusion.
7. Please rephrase this sentence, as it is beyond the scope of the results: "Widespread use can support optimisation of the MASLD clinical pathway and improve long-term patient management."

Outcomes:

"The primary outcome for this trial was to compare the difference in the proportion of patients with suspected MASLD incurring liver-related hospital consultations or liver biopsies between the study arms, from the date of randomization to the end of the study follow-up²⁰"

8. Please report the results of this primary outcome and clearly indicate it as the primary outcome, in the abstract and body of the paper. This is found on line 330 p12- please report there was no significant difference between arms.
9. Please report all secondary outcomes, and indicate they are secondary outcomes.

"the secondary outcomes included assessment of patient feedback from a patient satisfaction questionnaire, certainty of diagnosis, time to diagnosis, frequency of diagnosis, resource use (measured as rates of liver related outpatient investigations/ consultations/ hospital admissions during the study), cost effectiveness of the use of mpMRI (based on

randomisation arm comparison) and Personnel skills required for diagnosis (assessed as Percentage of total consultations performed by a specialist, at each specialist medical category, from date of randomisation to end of study)”

10. When reporting additional outcomes, please indicate that they are post hoc exploratory outcomes. This includes, for example, the stratification of patients as low or high risk, and the rate of “avoidable biopsy”, which are defined in the statistical analysis paragraph (p9 line 260) These ad hoc outcomes should not be reported in the abstract.

11. Please do not exclude patients from the analysis. (p 9 line 269 ff). Patients who did not achieve a useful image should be included in the denominator of all primary and secondary outcome comparisons for the imaging arm. These comparisons should be as randomized.

“rate of diagnosis (%-with-diagnosis) was assessed as the proportion of patients with a final diagnosis by the end of the trial follow-up period. “

12. Rate of diagnosis should use all randomized patients as the denominator, in each arm (an intent to treat analysis). Please report the underlying data in a table (can be supplementary), not just the percents in the figure (Fig 2). Figure 2 does not seem to support the claim in the abstract.

Version 4:

Reviewer comments:

Reviewer #7

(Remarks to the Author)

Previous comments by reviewer 6 have been addressed.

Reviewer #8

(Remarks to the Author)

The authors have done a superb job answering the statistical reviewer's comments and should be commended on their thoroughness. No further comments.

Reviewer #1 (Remarks to the Author):

Thank you for the opportunity to review this paper.

The introduction is fantastic and really lays the groundwork very well. The only comment here is about cT1. The introduction would benefit from some additional lead in regarding this characteristic (how its calculated/what it is/how mpMRI allows you to calculate cT1). It may be a very simple addition but since it is a modality of importance to this paper, a bit more background would be nice.

For what its worth, the real world nature is refreshing.

We would like to thank the reviewer for their very positive and kind comments. Below is a point-by-point response to the reviewers' comments.

The only comment here is about cT1. The introduction would benefit from some additional lead in regarding this characteristic (how its calculated/what it is/how mpMRI allows you to calculate cT1). It may be a very simple addition but since it is a modality of importance to this paper, a bit more background would be nice.

Thank you for your suggestion. We agree with the reviewer that we kept the introduction quite short and the text around cT1 quite light. We have now added the text below to the introduction. With regards to "how its calculated/what it is/how mpMRI allows you to calculate cT1" we have added a reference to Bachtiar et al 2019 which describes the methods used and have also added supplementary table S3 with the MRI parameters for acquiring mpMRI.

"Multiparametric magnetic resonance imaging (mpMRI), a standardised non-invasive non-contrast technology that uses post-processing of MRI images⁹, directly characterises disease activity (using corrected T1 [cT1] a metric of disease activity including fibro-inflammation), liver fat (using proton density fat fraction [PDFF]), and liver iron (using T2*)¹⁰⁻¹²."

Page 9, line 316, what does AI refer to? This paragraph is written a little wordy and hard to follow. Recommend reframing the results by study arm.

Thank you for your comments. It seems there is a typo in this section, and AI was meant to read IA. We apologise for our oversight, and we have edited the text throughout the manuscript and addressed all such typo's. We understand that this paragraph is a bit wordy. We have separated the findings by study arm and the text reporting the findings from SoC are now in their own paragraph:

"Of those recruited into the SoC arm, 23 patients (6%) had a liver biopsy. Of the 23 biopsies performed 11 (48%) patients did not have MASH with fibrosis following fibrosis testing."

Section 3.5: diagnosis of what? NAFLD, NASH, NASH with fibrosis? Also with nomenclature change, this would be a good time to mention that you may want to change to MASLD, MASH terminology.

Thank you for your comments. We have now amended all the terms to align with the new nomenclature change. For section 3.5, as shown in Fig 1A, patients were diagnosed of either NAFLD, NASH or NASH with fibrosis, NAFL/NASH with mixed aetiology, etc., and so to avoid having a lengthy sub-header, we had left it generic.

However, to address your comment, we have added the following sentence to the beginning of this section 3.5:

“In this study, as shown in Fig 1, patients were diagnosed with either MASLD, MASH or MASH with fibrosis, MASL/MASH with mixed aetiology or other aetiology, therefore, a diagnosis was defined as having a confirmed record of this by the end of the follow-up period.”

Page 11, line 384: "clinician the proportion"? something is incorrect here.

Thank you for your keen observation. Indeed, there seems to be a grammatical error in this sentence. It has now been edited and reads as follows:

“Secondly, **the proportion of patients** with a final diagnosis by the end of the trial follow-up period was significantly higher in the imaging arm compared to SoC alone implying that mpMRI has a significant impact on rate of diagnosis.”

Why did IA had less consultations/follow up with specialists than SoC? Is the idea that mpMRI should be used by PCPs/general practitioners but then NOT refer to hepatology when there is MASH +/- fibrosis. This doesn't add up. Furthermore, not every recruited patient came from a primary care pathway, so how is it that they did not see more specialists/clinical consultations? This part is a little unclear to me.

Thank you for your comment. The reviewer is indeed correct, although mpMRI has utility to rule-out MASH +/- fibrosis, in this study, our intention is not for PCPs/GPs to use mpMRI to avoid referring patients to hepatology (as patients were recruited from different care pathways), but rather to show the incremental benefit mpMRI can have on supporting decision making and thus reducing time to diagnosis.

From the findings obtained in this study, apart from the specialist at hospital visits (which included the MRI scan visit), patients in the SoC had more frequent visits to the GP, specialist outside hospital as well as the therapist. Moreover, there were significantly higher number of patient procedures (e.g. ultrasound testing) in the SoC arm, all of which resulted in more consultations/follow-up with specialists in the SoC. Therefore, as noted in the discussion section of the manuscript, it is possible that “*the use of mpMRI provided additional actionable clinical information to support patient clinical management which resulted in significantly fewer*” visits to specialists.

Many believe MASH is a biopsy proven diagnosis. Thus understanding how cT1 makes the diagnosis of MASH is extremely important here. Somehow this was only mentioned in the introduction.

Thank you for your comment. We have now amended the discussion as shown below:

“cT1 has shown clinical utility to support MASH diagnosis by enabling better risk assessment of high-risk MASH (at-risk MASH)^{17,33}, identifying those most likely to experience adverse outcomes^{15, 26,15} and in monitoring treatment responses in those with MASH^{8, 17, 34} in therapies with different mechanisms of action... Currently, cT1 is the first biomarker to have an accepted qualification plan in the FDA BQP for approval as a diagnostic enrichment biomarker in MASH clinical trials³⁵.”

Reviewer #2 (Remarks to the Author):

Shumbayawonda et al. describe a study investigating the cost-effectiveness of performing multiparametric MRI to evaluate patients with suspected non-alcoholic fatty liver disease (recently renamed to metabolic dysfunction associated steatotic liver disease, MASLD). A large, multi-centre, multi-national trial was conducted for this, including centres from Portugal, Germany, the Netherlands and the UK, including 802 patients. The authors found that compared to the standard-of-care patient pathway, patients undergoing mpMRI had fewer specialist appointments and patient assessments, as well as a higher proportion of them getting a definite diagnosis within a year. Whilst the short-term per-patient cost of including mpMRI in the patient pathway was higher than for standard-of-care, the authors argue that this difference will pay off in long term patient treatment/monitoring/management costs.

We thank the reviewer for a good summary of our work. Below is a point-by-point response to the reviewers' comments.

I only have a few small comments for the authors:

1. Please change NAFLD/NASH to MASLD/MASH throughout the paper to conform with the new nomenclature.

We thank the reviewer for their comment. Following the new nomenclature NAFLD/NASH has been changed to MASLD/MASH throughout the paper.

2. Histology is used to stage fibrosis, not to score it. Please replace references to "fibrosis score" with "fibrosis stage" throughout the manuscript.

We thank the reviewer for their comment. We have now edited the manuscript and all instances referring to 'scoring' have been changed to 'staging'.

3. Please include patient numbers referred for specialist appointments as well as patients with a diagnosis in the two groups - if this doesn't fit within the word limit of the manuscript, report them at least in the supplement

Thank you for your suggestion. To align with your comment below (comment 7), we have now edited table 2 to include the number of patients in the header. Furthermore, the proportion of patients with a diagnosis in each group has been reported both in section 3.5 and Fig 1.

4. P4L144: I suggest changing " ≥ 3 " to "3 or more" to help clarity

We thank the reviewer for their suggestion, we have made this edit to the manuscript text and it now reads as follows:

"For inclusion patients had to have either elevated liver enzymes, imaging suggestive of fatty liver disease, or have the presence of 3 or more of the following criteria..."

5. Please include sequence parameter details of the mpMRI protocol or refer to a previous study describing these. A more detailed description of the image analysis part is also required.

Thank you for your suggestion. We have now edited section 2.4 to include a reference to previous descriptions of the mpMRI protocol. Moreover, we have

added Supplementary table S3 which details the sequence parameters used in the mpMRI protocol.

“Supplementary table S3: MRI parameters for acquiring mpMRI”

Field of view (mm ³)	440 x 330 x 100
Reconstruction voxel size (mm)	1.15 x 1.15
Slice thickness (mm)	8
Slice gap (mm)	7
Slices	5
Parallel imaging SENSE factor	2
Repetition time (ms)	2.42
Echo time (ms)	1.05
Flip angle (°)	35
Acquisition duration	60 seconds
Respiratory compensation	5 breath-holds

6. Were the MR imaging read by radiologists for incidental findings or was mpMRI the only exam performed?

We thank the reviewer for their comment. All MRI scans were read for incidental findings following local site SoP. In this study, no additional incidental findings were identified following the addition of the MRI scan. To ensure this point is addressed, we have added the following text to section 2.4:

“All MRI scans were read for incidental findings following local site standards of practice.”

7. I suggest that the authors add the number of patients in the header of Table 2.

Thank you for your suggestion. Following on from your previous comment (comment 3), we have now edited table 2 to include the number of patients in the header.

Reviewer #3 (Remarks to the Author):

The manuscript by Shumbayawonda et al, 'Multiparametric MRI increases rate of NAFLD diagnosis with fewer specialist appointments and interventions: a real-world multi-national randomised trial' presents an analysis of the RADICAL-1 clinical trial, concluding that sole reliance on multi-paramateric MRI in comparison to 'Standard of Care' would reduce cost, specialist consultations, and unnecessary biopsies. It is laudable and no small feat to carry out a multi-national clinical trial across 10 different sites. However, there are a few points, that if clarified, could help with the clarity and confidence of the reported conclusions.

We would like to thank the reviewer for a good summary of our work and their kind comments. Below is a point-by-point response to the reviewers' comments.

As a point of clarification, obtaining an MRI is considered standard of care in workup of patients suspected NAFLD and the studies are typically done with some type of post-contrast T1 sequence as well as a sequence to assess fat content (PDFF in this trial). I could not find it explicitly stated, however does the SoC MRI not include these two sequences? Is the purpose of this study then to assess inclusion of cT1 and PDFF as part of the imaging workup?

Thank you for your comments. Although most sites do use MRI in various capacities, it is not universally adopted as part of SoC for the work-up of patients suspected MASLD (as noted in the current clinical guidelines: Rinella et al 2023). We agree with the reviewer that most sequences do typically use some type of post-contrast T1 sequence, however, the sequence used in this study is a distinct non-contrast and uses fast T1 acquisition to shorten the acquisition time (can be as short as 3 minutes, such as acquired as part of the UK biobank database (<https://www.ukbiobank.ac.uk/>)). To support clarification, section 2.4 has been edited to include a reference to the protocol used, and we have added Supplementary table 3 which details the sequence parameters used in the mpMRI protocol. As described in section 2.4, the sequence mentioned does include PDFF, alongside corrected T1, and T2*. The reviewer is correct in their assumption, SoC MRI does not typically include all three sequences in the same scan session.

Concerning the purpose of the study, the reviewer is indeed correct, as noted in the introduction, the "aim of this study was to evaluate the utility and cost of using mpMRI to risk stratify adults with suspected NAFLD". Therefore, part of our aim was to assess the inclusion of this distinct quantitative MRI procedure as part of the imaging workup for patients with MASLD.

Table 2 lists the total cost of MRI as €97,057.07 with mpMRI as an additional €101,148.06. Does this mean that the addition of those two sequences more than doubled the MRI cost?

We thank the reviewer for their comment and keen observation. Typically, the inclusion of a new technique into a clinical pathway, though cost effective, can be cost incurring at the start. It is worth noting that we did not perform two examinations, i.e. these are not two different sequences. The MRI cost reported herein is the scan time cost whilst that for the mpMRI is the cost associated with the post-processing and report delivery. Although in this study we separated the costs of MRI scanning and the mpMRI cost, it is envisaged that, similar to the adoption of other MRI protocols in real-world clinical practice, these costs will not be de-coupled, but rather, will be packaged as some kind of bundle cost. Nevertheless, for transparency's sake, we report herein the individual costs.

The reviewer is correct in their assumption, however, similar to the biopsy costs where costs associated with complications were not included, we only included the MRI costs for the scan time, thus, no associated costs e.g. clinician time, radiology time, assessment for incidental findings, and other overheads were included in the MRI costs presented herein. As noted above, the mpMRI cost are the costs for the post-processing, analyst time used, review and report delivery. Therefore, the costs presented for the MRI do not fully represent the total costs incurred by the imaging centre for conducting an MRI scan.

To address this comment, we have edited the text in section 4, and it now reads as follows:

“Similarly, when considering the MRI costs, only the costs for scan time was included for assessment, therefore no costs associated costs (including clinician time, radiology time, assessment for incidental findings, and other overheads) were included in the MRI costs presented herein.”

How was the gold standard for NAFLD defined? Histopathology is typically the definitive means. Line 241 of the manuscript has the first use of the NAS abbreviation -- this presumably refers to the NAFLD Activity Score -- however it is never defined (should be defined in text and added to abbreviation list). It is not clear how the NAS could be defined if biopsies were never obtained in the vast majority of cases.

Thank you for your comments. We have now edited the text and have added the definition of NAS to the text and the list of abbreviations.

According to the recent clinical guidelines, MAFLD diagnosis can be given where the patient has any one of five criteria mentioned in Rinella et al 2023. For MASH, latest clinical guidelines do not explicitly mention the requirement for liver biopsy for diagnosis as this procedure is not typically performed. For instance, in the international study reported by Ratziu et al 2022), in real-world management of patients, monitoring practices are not aligned with current guidelines and liver biopsy is not typically used for diagnosis.

With regards to the reviewers comment on the NAS, in section 2.6, when assessing the possibility for biopsy avoidance, this score was only used in the sub-group of patients who had a liver biopsy. In these cases, the NAS and fibrosis scores were reported and available for analysis as presented in section 3.3. with regards to using the NITs (e.g. cT1) to identify patients with high-risk disease, in addition to the text provided in section 1, we have also added the following in section 4:

“cT1 has shown clinical utility to support MASH diagnosis by enabling better risk assessment of high-risk MASH (at-risk MASH)^{17,33}, identifying those most likely to experience adverse outcomes^{15, 26,15} and in monitoring treatment responses in those with MASH^{8, 17, 34} in therapies with different mechanisms of action.”

What was the initial target enrollment for the study?

We thank the reviewer for their comment. This study aimed to recruit 1072 patients considering a 25% drop-out rate (as described in Tonev et al 2020). We have now added the following text to section 2.6:

“2.6 Sample Size Calculation

In a study by Blake et al¹⁶, the use of LiverMultiScan was found to result in a decrease in and 18% decrease in the use of biopsy. Adopting a conservative target of identifying a 14% decrease across different regions, each randomization arm is required to have 402 patients to maintain statistical significance with more than 80% power and show a difference in proportion of patients having consultations between the 2 arms. Due to the size of the trial, assuming there is a 25% dropout rate a total cohort of 1072 patients with will need to be recruited into the trial¹⁸.”

Statistical analysis states "between group", however cost differences between different countries is significant. It seems that it may have been more appropriate for comparison between the two arms for on a country by country basis. When looking at these differences, there are very different trends for each country, likely attributable to different implementations of health care models.

Thank you for your comment and keen observation. We agree with the reviewer that the cost differences between countries are significant and are likely due to the different implementations of health care models across the different countries. For instance, as shown in table 3, specialist appointments cost ~8.5 times more in The Netherlands compared to Portugal. Currently, there is no consensus on how to best make use of large samples of economic data when healthcare systems and pricing across jurisdictions vary. To reduce complexity in the main text, we report between group analyses (Fig 2), however, we have also added the comparisons between the two arms on a country-by-country basis to Supplementary Fig 2. Although there were numerical cost differences between the study arms, these did not reach statistical significance.

The protocol states that 1.5 or 3T scanners were used; there can be significant differences for cT1 and PDFFF obtained from these. How was variability for these differences accounted for?

We thank the reviewer for their comment and keen observation. The mpMRI protocol used has been standardised across scanner (Philips, GE, Siemens) and field strength (1.5 T and 3T) as reported by Bachtiar et al 2019. To support clarification, section 2.4 has been edited to include a reference to the protocol used (as reported by Bachtiar et al 2019), and we have added Supplementary table 3 which details the sequence parameters used in the mpMRI protocol.

“Supplementary table S3: MRI parameters for acquiring mpMRI”

Field of view (mm ³)	440 x 330 x 100
Reconstruction voxel size (mm)	1.15 x 1.15
Slice thickness (mm)	8
Slice gap (mm)	7
Slices	5
Parallel imaging SENSE factor	2
Repetition time (ms)	2.42
Echo time (ms)	1.05
Flip angle (°)	35
Acquisition duration	60 seconds
Respiratory compensation	5 breath-holds

I don't follow the logic for how an imaging measurement with a PPV of 0.75 and NPV of 0.83 could have supplanted any histopathological characterization in such a small sample size.

Thank you for your comment. With respect to biopsy avoidance, we agree with the reviewer that the sub-group of patients who underwent a liver biopsy in this study was small (albeit reflective of clinical practice (Ratziu et al 2022)), however, as stated in sections 2.6, following clinical guideline recommendations patients who do not have MASH with fibrosis should not be biopsied. Specifically, evidence available in literature (e.g. reported in Harrison et al 2018, Harrison et al 2020, Schaapman et al 2020, Jayaswal et al 2020, Imajo et al 2021, Andersson et al 2021, Roca-Fernandez et al 2023, etc.) shows the utility of mpMRI to identify high risk patients who should be biopsied. Thus, the results presented in section 3.3. are an assessment of the mpMRI markers (cT1 and PDFF) using thresholds reported in literature (using biopsy paired data), to identify the cases where mpMRI could have played a supporting role in ruling out those who do not require a biopsy non-invasively.

With regards to the reviewers comments about PPV and NPV, as noted by Mehta et al 2009 and also in the most recent Blue Cross Blue shield medical policy update (<https://www.bcbsm.com/amslibs/content/dam/public/mpr/mpresearch/pdf/2055449.pdf>), the high rate of sampling error in liver biopsy can lead to underdiagnosis and will bias estimates of performance characteristics of the noninvasive tests to which it is compared. From the analyses presented by Mehta et al 2009, the underperformance of liver biopsy will limit the perceived effectiveness of alternative technologies so much so that if the accuracy of biopsy is presumed to be 80%, a comparative technology with an AUROC curve of 0.76 may actually have an AUROC curve of 0.93 to 0.99 for diagnosing true disease. Although we do not report AUROCs in this study, it can be argued that an NIT with a PPV of 0.75 and NPV of 0.83 does have good utility to support liver biopsy avoidance.

Minor grammatical/typographical errors:

Line 384

Table 1: T2DB presumably should read T2DM

We thank the reviewer for spotting these errors. We have now edited the text accordingly.

Reviewer #4 (Remarks to the Author):

The authors should be commended for delivering a pragmatic real-world multinational randomised trial of multiparametric MRI in the evaluation of patients with NAFLD compared to standard of care (which was institution specific and not defined) in challenging circumstances with follow-up being curtailed because of the pandemic.

We thank the reviewer for their kind comments. Below is a point-by-point response to the reviewers' comments.

However, there are a number of significant methodological concerns in the paper. Different health economic evaluations are mentioned and appear to be used interchangeably despite being radically different things. The abstract starts off by stating that the study team investigated the utility and cost of mPMRI (utility is a very specific definition in health economics and is not described any further in the body of the article).

Thank you for your comment. We agree with the reviewer, in health economics, utility (typically reported as a number between 0 and 1) is a measure of the preference or value that an individual or society gives a particular health state. Herein, we evaluated clinical utility of mpMRI and not the health economic utility. To avoid confusion between the two, we have edited the manuscript where relevant and all instances where 'utility' refers to 'clinical utility' have been changed accordingly.

The primary objective of the study was to measure the cost-effectiveness of mPMRI but no cost-effectiveness analysis is presented. At best, a cost-minimisation analysis is described (though not explicitly defined as such and is not clearly articulated).

We thank the reviewer for their comments and suggestions. We agree with the reviewer that we did not include specific cost effectiveness analysis within the manuscript. We have added the following text to sections 2.5, 3.7 and 4:

It is worth noting that, in this study, in line with definitions and thresholds reported in literature²⁹⁻³¹ an incremental cost-effectiveness ratio (ICER) of \leq £20,000 per QALY gained has been considered as cost-effective in this population. To take into account variation in the difference healthcare systems, a conservative approach was used and thus, an ICER of \leq £10,000 (\sim €11,200) per QALY gained was considered as cost-effective in this study."

“3.7 Cost-effectiveness of associated with the use of mpMRI in adults with suspected MASLD

Considering the total costs for the IA (€522,473.63) compared to the SoC (€331,393.85) and the difference in the proportion of patients with a final diagnosis (%-with-diagnosis) (57% and 48% respectively), the ICER across the cohort was €5028.42. Using a QALY loss of 0.03 per patient per missed diagnosis, the use of mpMRI in the IA was cost-effective with an ICER of €5028.42 per 1.02 QALYs gained (€4929.82/QALY gained)."

“Sequential testing algorithms have been proposed to support better patient diagnosis and management of MASLD/MASH^{19,32}. For instance, the National Institute for Health and Care Excellence, which provides expert evidence-based best practice for health and care practitioners and whose guidance is targeted at ensuring the best care is delivered to patients in a fast manner whilst ensuring value for the taxpayer, encourages the uptake of best practice to improve outcomes. In their assessment^{29,30}, NICE considered an ICER of \leq £20,000 per QALY gained as cost effective. Although

a conservative approach was pursued in this study, with an ICER of \leq £10,000 per QALY gained considered as cost effective, the results indicate that the addition of mpMRI to the management plan to support disease (MASLD/MASH) diagnosis provides incremental gains of QALYs over SoC. By supporting diagnosis of significantly more patients, with the imminent regulatory approval of therapies to treat MASLD/MASH, early diagnosis of patients has the potential to limit the transition of patients towards more severe clinical outcomes (such as liver cirrhosis and end-stage liver disease) and their associated quality-of-life and economic costs³¹."

Another major limitation is that the field of NITs has moved dramatically since the timeline described in the trial. In the current era of expanded step-wise use of NITs, any potential cost saving with mPMRI in the field of MASLD is likely to be extremely limited or non-existent.

Thank you for your comment. We agree with the reviewer, the field of MASLD has experienced significant growth in recent years, with the use of NITs to support patient management carrying significant focus. However, although various publications from large consortia (e.g. LITMUS and NIMBLE) are emerging, where sequential testing of patients is being recommended to improve patient management and triaging, clinical guidelines are still calling for the identification of NITs to further support clinical management of patients with MASLD/MASH. Moreover, it is worth noting that patients with MASLD can be identified in multiple different pathways (not just from hepatology), and thus, the incremental cost saving associated with using NITs like mpMRI is neither extremely limited nor non-existent, but rather non-trivial. Thus, in the real-world, due to the high number of patients and the multiple branches of patient management, trade-offs between the number of tests used (including the cost of healthcare resource utilisation) and the time to diagnosis will likely inform cost savings associated with the use of different combinations of NITs in sequence.

Currently, patient pathways are recommended in clinical guidelines, however, as noted in many studies (e.g. Ratziu et al 2022), in real-world management of patients, monitoring practices are not aligned with current guidelines. Nevertheless, studies in literature, such as that performed by Blake and colleagues (2016) (albeit in a UK NHS setting), have evaluated the costs of using mpMRI in sequence following other tests and found that the inclusion of multiparametric MRI, in the diagnostic pathway of NAFLD may lead to cost savings.

Reviewer #5 (Remarks to the Author):

In the study titled “Multiparametric MRI increases rate of NAFLD diagnosis with fewer specialist appointments and interventions: a real-world multi-national randomised trial” Shumbayawonda Et Al. conducted a randomized trial with the primary objective of evaluating the utility and cost of using mpMRI to risk stratify adults with suspected NAFLD in a multinational setting. The paper is well written.

We would like to thank the reviewer for a good summary of our work and their kind comments. Below is a point-by-point response to the reviewers’ comments.

However, the methods and results had multiple major problems that potentially could impact the overall conclusion.

Introduction

- Authors should briefly recognize the NAFLD/NASH new nomenclature in their disease definition.

We thank the reviewer for their comment. Following the new nomenclature NAFLD/NASH has been changed to MASLD/MASH throughout the paper.

- Lines 107 to 109, the authors stated the following: “Studies have demonstrated the improved ability of cT1 in identifying and stratifying disease when compared to other noninvasive technologies”¹. The study cited was done in a Japanese cohort, which should be stated. In addition, the authors need to expand on what they mean by “disease” and which noninvasive technologies were investigated.
 - 1. Imajo, K., Tetlow, L., Dennis, A., Shumayawonda, E., Mouchti, S., Kendall, T.J., et al. (2021) ‘Quantitative multiparametric magnetic resonance imaging can aid non-alcoholic steatohepatitis diagnosis in a Japanese cohort’, World Journal of Gastroenterology, 27(7), pp.609-623. doi:10.3748/wjg.v27.i7.609

Thank you for your comment and keen observation. Included in the sentence noted by the reviewer are other papers highlighting the utility of the technology in MASH. Moreover, included in the manuscript are various publications (e.g. Harrison et al 2018, Imajo et al 2021, Jayaswal et al 2020, Harrison et al 2020, and Andersson et al 2021) reporting on the performance of cT1. cT1 is currently used in over 30 NAFLD/NASH clinical trials recruiting patients internationally. Furthermore, in addition to literature showing utility in a Japanese population, the mpMRI markers are included in various large consortia (e.g. LITMUS, UKBB and Dallas Hearts and Minds Study) which recruit patients across Europe and in the USA. Nevertheless, we agree with the reviewer that the use of only one reference following this part of the statement does not adequately represent the body of evidence available using this marker. We have now edited the text and added more references from non-Asian populations to support this statement.

With regard to lines 107-109, in this context, disease was implied to mean steatotic liver disease. We agree with the reviewer that in this sentence this definition is not very clear, we have now edited the text as shown below. Furthermore, to clarify that there are a range of noninvasive technologies compared to, we have added further text to address the reviewers comment:

“Studies have demonstrated the improved ability of cT1 in identifying and stratifying steatotic liver disease when compared to other noninvasive

technologies^{8,13,14} (including vibration controlled transient elastography [VCTE], magnetic Resonance Elastography [MRE] and shear wave elastography [SWE]), in predicting clinical outcomes^{14,15} and in monitoring and detecting early responses to treatment in those with MASLD/MASH^{16,17}.”

Methods

- In line 148, the authors stated the following in the exclusion criteria “alcohol over-use/abuse and any other cause of disease”. What disease are they referring to?

Thank you for your comment. Here, as noted in section 2.2, the disease referred to is any disease that the patient may have which in the opinion of the investigator, may affect the participant’s ability to participate in the study. For instance, patients with claustrophobia may not be able to tolerate having an MRI scan, and thus will be excluded from the study. To further clarify this statement however, we have added the following text:

“...which in the opinion of the investigator, based on medical records, may affect the participant’s ability”

- Power analysis needs to be included as part of the statistical analysis section

We thank the reviewer for their comment. The sample size calculation has already been described by Tonev et al 2020. Nevertheless, to address the reviewers comment, we have added further text to section 2.2 noting this and have added section 2.6 with the sample size calculation.

“The full study design and statistical powering for this trial has been reported by Tonev et al¹⁸ and full inclusion and exclusion criteria are shown in Supplementary table S1.”

“2.6 Sample Size Calculation

In a study by Blake et al¹⁶, the use of LiverMultiScan was found to result in a decrease in and 18% decrease in the use of biopsy. Adopting a conservative target of identifying a 14% decrease across different regions, each randomization arm is required to have 402 patients to maintain statistical significance with more than 80% power and show a difference in proportion of patients having consultations between the 2 arms. Due to the size of the trial, assuming there is a 25% dropout rate a total cohort of 1072 patients will need to be recruited into the trial¹⁸.”

- Case wise deletion assumes Missing Completely at Random (MCAR), which is unlikely to hold in this study. Proportions missing and their predictors need to be investigated, and imputation needs to be done as a sensitivity analysis. Otherwise, more justifications need to be provided for the use of a case-wise deletion strategy.

Thank you for your comment and suggestion. We agree with the reviewer, what was implied here was that in the imaging arm, all cases with missing MRI data were not included in the statistical analysis. For all other metrics reported (blood biomarkers, follow-up visits, etc.) all values were available for analysis. To address your comment, the text has now been edited as follows:

“Cases without reported MAS and fibrosis stage data were excluded from investigations looking at biopsy avoidance. Similarly, cases with missing MRI data due to the presence of imaging artefacts (including motion artefacts

[involuntary/voluntary patient motion], fluid heterogeneity [incorrect shim placement]), high liver fat outside the cT1 quantifiable range [PDFF>35%, or claustrophobia) were excluded from the analyses in the IA.”

- Given the missing patterns in the data and differences in SoC access sites, it would be more appropriate to examine the rate of use per patient. In addition, all outcome analyses should adjust for sites.

We thank the reviewer for their comment. Currently, there is no consensus on how to best make use of large samples of economic data when healthcare systems and pricing across jurisdictions vary. To support better understanding of the findings from this study, we have reported results on the cohort level (aggregated across all sites), country level (from each clinical site), and have also added mean per patient costs (both in aggregate and per country). We do agree with the reviewer that it could be helpful to examine the rate of use per patient, however, to reduce complexity in reporting and summarising the findings from this study, it was our view that reporting the rate of use per patient may not be applicable where care pathways and management styles differ. Hence, when taking into account different trends for each country, variation in costs, and the different implementations of health care models, it was our view that examining the aspects presented in the manuscript would provide a more holistic view of the findings.

Results

- Study design was 1:1; however, the SoC group had 4 patients less than the intervention arm. An explanation for this needs to be included in the results section 3.1. More importantly, a full description of the final analysis sample needs to be included here. According to the

Thank you for your comment and suggestions. Indeed, the study was a 1:1 designation of patients to a study arm. More specifically, patients were assigned to a study arm after entering their details into the study electronic data capture. Therefore, due to patient withdrawal following enrolment from the study at some sites a minor difference of 4 patients between the study arms was observed in the final cohort.

The sample size calculation has already been described by Tonev et al 2020. Nevertheless, to address the reviewers comment, we have added further text to section 2.2 noting this and have added section 2.6 with the sample size calculation.

“The full study design and statistical powering for this trial has been reported by Tonev et al¹⁸ and full inclusion and exclusion criteria are shown in Supplementary table S1.”

“2.6 Sample Size Calculation

In a study by Blake et al¹⁶, the use of LiverMultiScan was found to result in a decrease in and 18% decrease in the use of biopsy. Adopting a conservative target of identifying a 14% decrease across different regions, each randomization arm is required to have 402 patients to maintain statistical significance with more than 80% power and show a difference in proportion of patients having consultations between the 2 arms. Due to the size of the trial, assuming there is a 25% dropout rate a total cohort of 1072 patients with will need to be recruited into the trial¹⁸.”

- To assess overall balance and potential difference in patient characteristics between participating sites, Table One should include p-values to support the claim of no statistically significant differences between 1) intervention and SoC groups and 2) overall difference between participating sites. Variables that are significantly different need to be accounted for in the outcome models.

We thank the reviewer for their comments and suggestions. We have now edited table 1 to include p-values for 1) intervention and SoC groups and 2) overall difference between participating sites.

- Tables must include the number of patients included in the analyses.

Thank you for your suggestion. We have now edited table 2 to include the number of patients in the header.

- Using simple arithmetic to determine the number of patients used for analyses in Table Two, it seems that the intervention arm had 345 patients while the SoC group had 365. If these numbers are correct, then using aggregate numbers is inaccurate. Instead of reporting total numbers, the rate of consultations should be estimated using poisson regression while accounting for the size of each site.

We would like to thank the reviewer for spotting this error in our calculations. It seems that there was an error during the compilation of this table, and we did not update this entry correctly. We can confirm that we have checked all analyses and numbers and the results provided in table 2 are indeed for the entire cohort, i.e. n=802 not n=710. Following all the comments raised around the numbers of patients used in table 2, to ensure uniformity in entries presented (total consultations/procedures/tests, total costs, and per patient costs) we have now edited the table accordingly.

- It also seems that table one does not describe the analytic sample used in table 2 given the differences in the numbers of participants in each group. Table Two was done on 710 participants, and table one describes 802. This is problematic in an intention-to-treat analysis.

Thank you for your follow-on comment. We apologise for this oversight on our part and the confusion this may have caused. As mentioned in the reply above, there was an error in the per-patient calculations for the individual sections of table 2, however, the overall per patient costs presented in the final section of the table are correct. We agree with the reviewer that errors of this type can be problematic in intention-to-treat analyses and have now reviewed all results presented in the tables to ensure they are accurate.

- General comment: Tables and figures titles should be limited to titles with no result interpretations. For example, the Table One title should be limited to "Table 1 – Overall and regional baseline demographic summary." The following should be part of the results "Despite variations in patient management, there were no significant differences in age, sex, weight, BMI, biochemical markers or proportion with T2DM observed between the two trial arms."

Thank you for your comments. We do agree with the reviewer that captions should be limited to only the relevant information. We have now edited all the table and figure captions accordingly.

Reviewers' comments:

Reviewer #1 (Remarks to the Author):

My comments have been addressed.

We are glad that the reviewer found our rebuttal sufficient. We would like to thank them for the time spent providing comments to improve our manuscript.

Reviewer #2 (Remarks to the Author):

I would like to thank the authors for amending their manuscript, which now reads really well.

We would like to thank the reviewer for their kind comments. Below is a point-by-point response to the reviewers' comments.

I only have two small comments:

1. Since the mpMRI protocol was used on all three major MRI vendors, I suggest the authors remove the reference to SENSE in Table S3, as GRAPPA is likely the acceleration method of choice on Siemens machines.

Thank you for your suggestion. We agree with the reviewer and have now edited table S3 to remove the reference to SENSE. The table now reads as shown below.

Field of view (mm ³)	440 x 330 x 100
Reconstruction voxel size (mm)	1.15 x 1.15
Slice thickness (mm)	8
Slice gap (mm)	7
Slices	5
Parallel imaging factor	2
Repetition time (ms)	2.42
Echo time (ms)	1.05
Flip angle (°)	35
Acquisition duration	60 seconds
Respiratory compensation	5 breath-holds

2. Please correct the reference errors in Table 3.

We would like to thank the reviewer for their keen observation. We have now corrected the references in Table 3 and the changes are shown below:

Site	GP	Therapist	Specialist	Blood	Ultrasound	MRI	Elastography	Biopsy	mpMRI
Coimbra (Portugal) € ²¹	4.5	16	31	6.71	19.52	127.9	25.7	285.6	301
Ulm (Germany) € ²²	59.45	59.45	112.84	94.77	92.67	470.38	100	131.1	301
Leiden (The Netherlands) €	15.7	24.5	269.81	5.0	105.51	425.8	100	732.85	301
UK £ ²³	132	45	279	5.0	39	130	49	726	258

Reviewer #3 (Remarks to the Author):

The authors have adequately addressed the concerns.

We are glad that the reviewer found our rebuttal sufficient and that we addressed their concerns. We would like to thank the reviewer for the time spent providing comments to improve our manuscript.

Reviewer #4 (Remarks to the Author):

Thank you for your response to my review comments. I am satisfied that these have been appropriately addressed.

We are glad that the reviewer found our rebuttal sufficient and that all their comments were appropriately addressed. We would like to thank the reviewer for the time spent providing comments to improve our manuscript.

July 17th 2024,

Dear Dr Amir Zarrinpar,

Re: COMMSMED-23-0693C

Thank you for the opportunity to further revise our manuscript following your comments pertaining to the handling missing data raised by the fifth reviewer. Please find attached, a detailed response to your query. All changes to the manuscript mentioned in the responses below have been highlighted in yellow for ease of identification.

With kind regards,

Elizabeth Shumbayawonda
(on behalf of the authors)

Editors comments:

The one specific point that needs further clarification is the approach to handling missing data raised by the fifth reviewer. Reviewer 5 previously raised concerns about the assumption that the missing data was Missing Completely at Random (MCAR). In your revision, you focused on the exclusion criteria rather than directly addressing this issue. We need you to provide a sensitivity analysis or a detailed discussion on the potential limitations of your study should the assumption about missing data not being random hold true. Please address this point comprehensively in your revised manuscript.

We would like to thank the editor their comment. Below we have a detailed response to your comment.

Firstly, we have added the comment from Reviewer 5 and the submitted rebuttal answer for the first round of review (COMMSMED-23-0693):

Reviewer 5 comment:

- *Case wise deletion assumes Missing Completely at Random (MCAR), which is unlikely to hold in this study. Proportions missing and their predictors need to be investigated, and imputation needs to be done as a sensitivity analysis. Otherwise, more justifications need to be provided for the use of a case-wise deletion strategy.*

Thank you for your comment and suggestion. We agree with the reviewer, what was implied here was that in the imaging arm, all cases with missing MRI data were not included in the statistical analysis. For all other metrics reported (blood biomarkers, follow-up visits, etc.) all values were available for analysis. To address your comment, the text has now been edited as follows:

“Cases without reported MAS and fibrosis stage data were excluded from investigations looking at biopsy avoidance. Similarly, cases with missing MRI data due to the presence of imaging artefacts (including motion artefacts [involuntary/voluntary patient motion], fluid heterogeneity [incorrect shim placement]), high liver fat outside the cT1 quantifiable range [PDFF>35%], or claustrophobia) were excluded from the analyses in the IA.”

Firstly, we agree with the Editor that we could have provided a more detailed response to Reviewer 5. We have now endeavoured to do so below.

When data are Missing Completely at Random (MCAR), their missingness is independent of the observed and unobserved data, i.e. no systematic differences exist between participants with missing data and those with complete data. In addition to this, in the assessment of missingness, the amount (number of variables) with missing data should also be considered. Where there is ~ 10% missing data, it is typically acceptable to use exclusion and not require sensitivity analysis i.e. MCAR holds. An assessment of missing data showed that in this study, of those who got:

- a liver biopsy – out of the 44 biopsies performed 2 had missing data (mainly components of the MAS score). Specifically, of the 21 biopsies performed in the imaging arm, these 2 cases with missing data were excluded and only 19 cases reported on (as noted in table 2 and section 3.3 Biopsy avoidance).
- an MRI scan – out of the 403 patients who were randomised to the imaging arm 21 had missing data (this includes the 7 patients who declined or were unable to have an MRI scan) (table 1).

Thus, although we agree that in some cases MCAR is generally regarded as a strong assumption, we believe that it does hold in this study and thus was not considered to be a significant cause of bias due to the low percentage of missing data. Therefore, considering the above, we do not believe that a sensitivity analysis is required as ~10% missing data is typically acceptable in healthcare analyses of this type. Nevertheless, to ensure that this is addressed in the manuscript we have added the following text:

To address the excluded cases with missing MRI data, the following text was added to *section: 3.1 Participant demographics, randomisation, and baseline characteristics* (lines: 302-304):

“Those with missing cT1 values, 5% (21), due to either the presence of imaging artefacts or high liver fat outside the cT1 quantifiable range, were excluded from this analysis.”

To address the excluded cases with missing liver biopsy data, the following text was added to *section: 3.3 Biopsy avoidance* (lines: 339-340):

“In the IA, 21 patients received a liver biopsy, but 2 had missing data and were not included in any analyses.”

To address the limitations this could have, the following text was added to *section 4: Discussion* (lines: 532-544):

“Data missingness can be a source of potential bias as it can reduce the size of the sample available for analysis. Alongside the overall assessment of data missingness, in cases where over 10% of data is missing across multiple variables, sensitivity analyses are typically required to evaluate potential impact on analysis results and account for sources of bias⁴⁰. In this study, 2 cases with missing histology data and 21 cases with missing imaging data (including those who declined or were unable to have an MRI scan) were excluded from some analyses. As this was within the acceptable range of allowable missing data in healthcare investigations (~10%), missing data were assumed to be missing completely at random (MCAR), and thus no additional sensitivity analyses were performed. It is worth noting that in some cases the causes of missing data cannot be addressed as they are themselves unmeasured and multifaceted. This can further be complicated in multi-institutional multi-national studies such as that presented herein. Nevertheless, future studies should make note of these limitations and the potential impact they could have on reported results⁴⁰.”

Referee expertise:

Referee #6: Biostatistics and Bioinformatics

Reviewers' comments:

Reviewer #6 (Remarks to the Author):

In general this seems like an important and well conducted study. However, the results might be presented more carefully. The paper should clearly report the primary outcome and all secondary outcomes of the trial, indicated as such. These primary and secondary comparisons should be of the as-randomized study population, without excluding any patients (such as those who did not get a high quality image or biopsy-they still count!).

Additional outcomes beyond the primary and secondary outcomes of the trial should be clearly indicated as post hoc and exploratory, and should not be reported in the abstract.

For example, the proportion of patients “incurring liver-related hospital consultations or liver biopsies” should include all patients, even those in the imaging arm without successful imaging. (This was the primary outcome). Similarly, the proportion of patients with a final diagnosis should include those patients who had a failed imaging study or failed biopsy.

Please double check that statements in the abstract are well supported by the data- this did not always seem to be the case.

Please limit statements in the conclusion to primary or secondary trial outcomes.

Thank you for your comments and review of our work. Indeed, we agree with the reviewer that all patients count in such analyses and have added the rates and percentages where applicable throughout the manuscript. To distinguish the primary outcome from the secondary outcomes, we have also made changes to make sure the text aligns with these points and edited the abstract text as necessary to ensure everything is aligned. Please find more specific point-by-point replies to your comments below.

Specific comments.

Abstract.

1. Please give effect sizes with 95% confidence intervals (e.g. percentage point differences in rates) rather than p values.

Thank you for your comment. Due to the variation in patient care across different centres and countries, and the differences in recruited patients between different centres, reporting findings as rates rather than frequencies may not convey the findings in a way that can be easily digestible and comparable between regions. For instance, the Netherlands and Portugal have very different patient management models. Taking therapist appointments in SoC as an example, the 88 patients in the Netherlands received 498 total consultations (~5-6 consultations per person) whereas for the 75

patients recruited in Coimbra only 3 received a therapist appointment. Thus, although we agree with the reviewer that reporting rates and 95% confidence intervals can be informative, as there is no standard approved way to report such data, in this instance, we believe that it may hinder effective comparison and understanding of the frequency of events with respect to the population.

Nevertheless, as reporting the rates may give better granularity as to the actual numbers of patients the frequencies pertain to, to ensure that we have incorporated the reviewer's comment, we have added the actual numbers alongside frequencies in the manuscript text. Sections amended:

3.1 Participant demographics, randomisation, and baseline characteristics

- "There was high patient acceptance of the MRI scan resulting with only 1.7% (7/403) declining or being unable to have an MRI scan."
- "In the IA, 66% (267/403) of patients had high liver fat (PDFF \geq 10%) or evidence of disease activity (cT1 \geq 800ms"
- "In this study, 5% (21/403) of patients in the IA had missing cT1 values, due to either the presence of imaging artefacts or high liver fat outside the cT1 quantifiable range"

3.2 Clinical attendances and investigations

- "Patients in the SoC arm (3479) attended significantly more (9%) specialist appointments (with healthcare professionals and for patient assessments) compared to those in the IA arm (3839, p=0.015) (Table 2)"
- "At the end of the trial, 53% (423/802) of the cohort had a final diagnosis (Fig 1). When comparing the two trial arms, despite having fewer patient appointments, the IA (57%; 230/403) had significantly higher %-with-diagnosis compared to SoC (48%; 193/399, p= 0.0012; Fig 2, Table 2)."

3.3 Biopsy avoidance

- "In the IA, 21 (5%, 21/403) patients received a liver biopsy, but 2 had missing data and 1 had comorbid autoimmune hepatitis."
- "Biopsy results showed that 8 patients (40%; 8/20) had MASH with fibrosis, while 10 patients (50%, 10/20) did not meet the histological criteria for this diagnosis and thus could have avoided having a biopsy (Fig 2)."
- "Hence, 9 (45%, 9/20) biopsies performed in this patient group, indicated as necessary following fibrosis testing and clinician discretion, could have been avoided using mpMRI"
- "Of those recruited into the SoC arm, 23 patients (6%, 23/399) had a liver biopsy. Of the 23 biopsies performed 11 (48%, 11/23) patients did not have MASH with fibrosis following fibrosis testing"

3.4 Differences in patient care

- "In terms of specialist referral, a reduction in the proportion of patients accessing specialist care (with healthcare professionals and for patient assessments) was seen in the IA in comparison to SoC, in Germany (19%; IA:1090 visit vs SoC:1344 visits) and UK (14%; IA:421 visit vs SoC:487 visits)."
- "Notably, patients in Portugal were managed by their general practitioner and relatively few patients (14.0% (11/79) and 9.0% (7/75) for IA and SoC

respectively) accessed liver-related specialist care (Table 2; Supplementary Fig S1).”

3.5 Differences in and rate of diagnosis

- “Overall, there was a higher %-with-diagnosis in the IA (57%; 230/403) compared to the SoC arm (48%; 193/399) and was significant in Germany and Netherlands (both $p < 0.001$, Fig 2).”
- “Conversely, in Portugal where patients were largely managed in primary care only 4% (3/79) of patients in the IA and 3% (2/75) in SoC had a confirmed diagnosis at the end of the follow-up period.”
- “In the UK, %-with-diagnosis was comparable in both the imaging and SoC arms (40% (51/127) and 44% (56/128) respectively; Table 2).”

2. Please report the primary outcome of the trial in the abstract and report all secondary outcomes. The primary outcome was not different between arms

Thank you for your suggestion. The primary objective for the study was to “investigate whether the introduction of mpMRI as a standardised diagnostic test for liver disease can prove a cost-effective method by reducing the number of patients with suspected MASLD who incur liver-related hospital consultations, including liver biopsies”, and thus the aim was stated as being the evaluation of “the utility and cost of using mpMRI to risk stratify adults with suspected MASLD in a multinational setting”. To ensure that the text in the abstract aligns with this, we have edited the abstract to report the primary findings (cost effectiveness) first followed by all the other secondary outcome findings. With respect to the reviewers’ comment that “The primary outcome was not different between arms” as presented in section **3.7 Cost-effectiveness of associated with the use of mpMRI in adults with suspected MASLD**, indeed the IA was more cost effective.

Findings: The inclusion of mpMRI was cost-effective with an ICER of €4968 per QALY gained. SoC had significantly more specialist appointments ($p = 0.015$) and patient assessments ($p < 0.001$). Across all involved hospitals, %diagnosis was significantly higher in the imaging arm ($p = 0.0012$). cT1 correctly classified 50% of patients who did not have MASH with fibrosis and could have avoided biopsy. Including all costs evaluated the imaging arm incurred higher short-term per-patient healthcare expenditure compared to the SoC arm (€1,300 vs. €830).”

3. Across all involved hospitals, %diagnosis was significantly higher in the imaging arm ($p = 0.0012$). I did not find that this statement was supported by the results (fig 2). The underlying data should be reported, and this analysis should use all patients, as randomized.

Thank you for your comment and keen observation. Firstly, as depicted in the figure, there was a significant difference in Fig 2A with a level of 2 stars (**). In the figure caption we have noted that ** indicates $p < 0.01$. For the number of procedures, these data are already reported in Table 2. As this statement is in the abstract, we cannot

add references to the manuscript tables, however, to ensure that the readers can also follow this finding, we have added the text below:

“At the end of the trial, 53% (423/802) of the cohort had a final diagnosis (Fig 1). When comparing the two trial arms, despite having fewer patient appointments, the IA (57%; 230/403) had significantly higher %-with-diagnosis compared to SoC (48%; 193/399, $p= 0.0012$; Fig 2, Table 2).”

Additionally, to ensure the reader can follow where the frequencies are derived from, we have added the number of patients who received a diagnosis to table 2.

Conclusion:

4. Please rephrase the conclusion in terms of the primary outcome of the trial and the cost effectiveness.

Thank you for your suggestions, we have now edited the abstract conclusion and it now includes the primary outcome. (see point 7 for all included edits)

5. Please rephrase “without any additional health care resource use” as “without increase in other liver-related health care resource use”

Thank you for your suggestion, we have made this change. (see point 7 for all included edits)

6. Please include the increase in cost in the conclusion.

Thank you for your suggestion, we have made this change. (see point 7 for all included edits)

7. Please rephrase this sentence, as it is beyond the scope of the results: “Widespread use can support optimisation of the MASLD clinical pathway and improve long-term patient management.”

Thank you for your comment, we have now edited the conclusion following all comments from points 4-7 and the conclusion now reads as follows:

“**Conclusion:** Addition of mpMRI to SoC for the management of adults with suspected MASLD multi-nationally was cost-effective, enhanced rate of diagnosis multi-nationally and increased rate of diagnosis without increase in other liver-related health care resource use. Due to the need for standardisation of SoC, widespread use can support optimisation of the MASLD clinical pathway and improve long-term patient management”

Outcomes:

“The primary outcome for this trial was to compare the difference in the proportion of patients with suspected MASLD incurring liver-related hospital consultations or liver

biopsies between the study arms, from the date of randomization to the end of the study follow-up²⁰”

8. Please report the results of this primary outcome and clearly indicate it as the primary outcome, in the abstract and body of the paper. This is found on line 330 p12- please report there was no significant difference between arms.

Thank you for your comments and suggestions. We agree that the primary outcome of the study was to assess the utility and cost-effectiveness of the inclusion of mpMRI in SoC, we have reported this in both the abstract and in the manuscript conclusion. We have also edited the first paragraph of the discussion to ensure that the summary of key findings are related to the primary outcomes. The text now reads as follows:

“Firstly, regarding the primary objective of this study, in a population with suspected MASLD, mpMRI is a cost-effective tool with an ICER of €4,929/QALY gained. Secondly, with regards to liver-related consultations, when using mpMRI as part of patient management, there are significantly less clinical consultations and follow-up appointments with healthcare specialists when compared to standard of care alone. Additionally, the proportion of patients with a final diagnosis by the end of the trial follow-up period was significantly higher in the imaging arm compared to SoC alone implying that mpMRI has a significant impact on rate of diagnosis.”

Looking at line 330 on p12 (now line 324 p11), there were significant differences in the %-with-diagnosis between those in the IA vs SoC arm. To support answering this comment, we have broken down the primary outcome:

- (1) **Reviewer comment: liver-related hospital consultations or liver biopsies between the study arms**
 - The findings for this are reported in sections 3.2, 3.4, 3.5 for consultations and 3.3 for biopsies respectively.
- (2) **Reviewer comment: on line 330 p12 please report there was no significant difference between arms**
 - As noted in detail in point 3 above, the differences between arms were significant and results can be found in section 3.2

9. Please report all secondary outcomes, and indicate they are secondary outcomes.

“the secondary outcomes included assessment of patient feedback from a patient satisfaction questionnaire, certainty of diagnosis, time to diagnosis, frequency of diagnosis, resource use (measured as rates of liver related outpatient investigations/ consultations/ hospital admissions during the study), cost effectiveness of the use of mpMRI (based on randomisation arm comparison) and Personnel skills required for diagnosis (assessed as Percentage of total consultations performed by a specialist, at each specialist medical category, from date of randomisation to end of study)”

Thank you for your suggestion. We have edited section 2.5 to ensure grouping of the primary and secondary objectives and outcomes, as well as to ensure that the reporting of the outcomes are consistent throughout the text. We had reported all the

secondary outcomes/objectives for the study, but these are not all those reported/explored in this work. For instance, the findings from the patient satisfaction questionnaire (EQ-5D-5L) pertain to patient quality of life (QoL) which has not been reported as it is a different subject to that reported in this manuscript. Thus, to avoid confusion for the reader, and to address the reviewers' comment, we have edited the text to report only those aspects which have been reported in this manuscript:

“The primary objective for this trial was to measure the cost effectiveness of LiverMultiScan as a standardised diagnostic test for liver disease in different EU territories. The primary outcome for this trial was to investigate the utility and cost-effectiveness of including multiparametric magnetic resonance imaging (mpMRI) to the management of adults with suspected MASLD incurring liver-related hospital consultations including liver biopsies between the study arms²⁰. It is worth noting that, in this study, in line with definitions and thresholds reported in literature^{29–31} an incremental cost-effectiveness ratio (ICER) of ≤£20,000 per QALY gained has been considered as cost-effective in this population. To take into account variation in the difference healthcare systems, a conservative approach was used and thus, an ICER of ≤£10,000 (~€11,200) per QALY gained was considered as cost-effective in this study. There were multiple secondary objectives for this trial including assessment of %-with-diagnosis, frequency of diagnosis and resource use (including access to specialist care) prior to MASLD/MASH diagnosis. Moreover, the secondary outcomes included assessment of certainty of diagnosis, frequency of diagnosis and resource use (measured as rates of liver related outpatient investigations/ consultations/ hospital admissions during the study)²⁰.”

For clarity, please see the additional breakdown below:

Primary outcomes:

- Cost effectiveness – section 3.7
- Differences in liver-related hospital consultations including liver biopsies – sections 3.2, 3.4, 3.5 (consultations) and 3.3 (liver biopsies)

Secondary outcomes:

- %-with-diagnosis – section 3.5
- Frequency of diagnosis and resource use – sections 3.4, 3.5
- Assessment of certainty of diagnosis, frequency of diagnosis – sections section 3.5

10. When reporting additional outcomes, please indicate that they are post hoc exploratory outcomes. This includes, for example, the stratification of patients as low or high risk, and the rate of “avoidable biopsy”, which are defined in the statistical analysis paragraph (p9 line 260) These ad hoc outcomes should not be reported in the abstract.

Thank you for your suggestions. The field of MASLD has changed significantly over the years. For instance, there has been a change in nomenclature recently (from NAFLD to MASLD) which has been followed by a change in diagnostic criteria.

However, irrespective of this change, the requirement for liver biopsy is a very relevant point for those managing metabolic diseases such as MASLD and MASH. Therefore, as the population of readers, who will encompass clinicians who manage these patients (e.g. Hepatologists, Gastroenterologists, Endocrinologists, etc.) will be interested to know this finding, reporting it is important. This importance was also acknowledged and noted in the comments by Reviewer 1 (Referee expertise: NAFLD/MASLD clinician with expertise in diagnostic testing) and Reviewer 3 (Referee expertise: MR imaging in NAFLD/MASLD). Thus, we believe that keeping the rate of “avoidable biopsy” in the abstract is relevant.

Regarding the biopsy avoidance and stratification of patients as low or high risk, these were not post-hoc analyses as they were included in the approved study protocol before the data were analysed following the end of the trial.

11. Please do not exclude patients from the analysis. (p 9 line 269 ff). Patients who did not achieve a useful image should be included in the denominator of all primary and secondary outcome comparisons for the imaging arm. These comparisons should be as randomized.

“rate of diagnosis (%-with-diagnosis) was assessed as the proportion of patients with a final diagnosis by the end of the trial follow-up period.”

Thank you for reiterating this comment. As described above in point 1, we have now amended all calculations and included all patients numbers to the reported figures. It is important to note however, that autoimmune hepatitis is an exclusion criterion for the diagnosis of MASLD/MASH as reported in clinical practice guidelines (PMID: 36727674, PMID: 38851997). Therefore, we excluded this patient from analyses.

12. Rate of diagnosis should use all randomized patients as the denominator, in each arm (an intent to treat analysis). Please report the underlying data in a table (can be supplementary), not just the percents in the figure (Fig 2). Figure 2 does not seem to support the claim in the abstract.

Thank you for your comment and suggestion. We have now edited the text as noted above in points 1 and 11 regarding the use of all patients within the denominator.

With specific reference to the findings reported in the abstract as well as in Fig 2, to the best of our understanding, we have reported on points which are supported by the outcomes in the Results section. Specifically, within the Findings section we report:

1. The inclusion of mpMRI was cost-effective with an ICER of €5521 per QALY gained – finding from section 3.7
2. SoC had significantly more specialist appointments ($p=0.015$) and patient assessments ($p<0.001$) - finding from section 3.2
3. Across all involved hospitals, %diagnosis was significantly higher in the imaging arm ($p=0.0012$) - finding from section 3.2
4. cT1 correctly classified 50% of patients who did not have MASH with fibrosis and could have avoided biopsy - finding from section 3.3

5. Including all costs evaluated the imaging arm incurred higher short-term per-patient healthcare expenditure compared to the SoC arm (€1,296 vs. €830) - finding from section - finding from section 3.2

Additionally, with respect to figure 2, the results shown here illustrate the below 3 findings.

1. The proportion of patients who received a diagnosis in the total cohort as well as in the individual territories (Fig 2A) – this indicates the %-with-diagnosis as reported in sections 3.2 and 3.5
2. The number of visits with health care practitioners (Fig 2B) – this indicates frequency of liver related hospital visits which are part of the study aims
3. Costs incurred in the standard of care management of patients in the imaging arm (without inclusion of MRI costs) and the SoC arm – this indicated the like-for-like costs associated in the imaging arm and the SoC arm. Costs including the MRI costs are reported in Supplementary Fig 2.

The actual numbers used to generate these plots (in Fig 2) can be found in table 2 and have also been included throughout the text as noted in the rebuttal to point 1 above. From our understanding, and as summarised in this answer, we have reported findings which are shown in the study outcomes of the paper. As mentioned in the response to point 3, we have now added the number of patients who has a diagnosis at the end of the trial to table 2. These numbers are the ones presented in Fig 2.